

# Unique structure, radiative effects and precipitation characteristics of deep convection systems in the Tibetan Plateau compared to tropical oceans

Yuxin Zhao[1], Jiming Li[1]*, Deyu Wen[1], Yarong Li[1], Yuan Wang[2], Jianping Huang[1]

[1]Key Laboratory for Semi-Arid Climate Change of the Ministry of Education, College of Atmospheric Sciences, Lanzhou University, Lanzhou, China

[2]Collaborative Innovation Center for Western Ecological Safety, Lanzhou University, Lanzhou, China

*Correspondence to*: Jiming Li (lijiming@lzu.edu.cn)

**Abstract.** Using the space-borne lidar/radar observations, this study identifies deep convection systems (DCS), including deep convection core (DCC) and anvils, over Tibetan Plateau (TP) and tropical oceans (TO), and finds that DCSs over TP are less frequent, exhibiting narrower and thinner DCCs and anvils compared to those over TO. The thinner DCCs over TP exert weaker radiative cooling effects at the top of atmosphere (TOA) compared to the TO. But, the shortwave TOA cloud radiative effect (CRE) of TP anvils is stronger than that of the TO possibly due to more densely packed cloud tops over TP. It results in stronger TOA CRE of DCSs over TP than that of TO. Especially, longwave CRE of DCSs over TP is notable greater at surface and low-level atmosphere due to the distinct lower temperature and less water vapor. The width of DCSs shows a positive correlation with wind shear and atmospheric instability, and the underlying mechanisms are discussed. We also find that the impact of aerosols on cloud top heights and precipitation displays significant discrepancies between the two regions. It is because that the aerosol invigoration effect is less efficient on the TP DCSs, mainly attributed to the significantly colder cloud base. Due to competition between invigoration and radiative effects of aerosols, the correlation between precipitation and aerosols over TP is not significant. However, precipitation in the TO experiences invigoration followed by suppression with increasing aerosols, due to the dominance of aerosol radiative effect and enhancement entrainment under polluted conditions.



## 1 introduction

The Tibetan Plateau (TP), has received global attention as the heat source of the East Asian summer monsoon due to its prominent altitude and special topography (Wu et al., 2017; Duan et al., 2020). Over the Tibetan Plateau, surface heating causes a low-pressure centre, which can attract the warm-moist air convergence from the ocean and then promote convection activities (Wu et al., 2012). It brings abundant water to the atmosphere over the Tibetan Plateau and its surrounding regions, and thus the Tibetan Plateau is referred to as the "Asian water tower" (Xu et al., 2008). Over the past decades, some studies have shown that the Tibetan Plateau has experienced significant climate change (Liu and Chen, 2000; Yao et al., 2012). The apparent climate change is closely related to the variations in cloud properties, such as cloud types (Duan and Wu, 2006). In fact, based on the ground-based observations and ISCCP data, Yang et al. (2012) have pointed out that the contradiction between the decrease in total cloud cover and solar radiation dimming over the Tibetan Plateau can be explained by the increasing deep convective clouds. Although the deep convective cloud is less frequent, it has a more complicated vertical structure and larger extent compared to other cloud types, thus exerts a great influence on radiation and precipitation over Tibetan Plateau region (Dong et al., 2016; Yan and Liu, 2019). In particular, the deep convective clouds developed over the Tibetan Plateau can move downstream and cause strong precipitation and storms in eastern China (Hu et al., 2016; Liu et al., 2019b; Chen et al., 2020). Additionally, some convections also can penetrate the tropopause and affect tropospheric-stratospheric matter exchange, such as sending water vapour into the stratosphere and changing stratospheric multi-material concentrations through chemical reactions (Kirk-Davidoff et al., 1999; Forster and Shine, 2002; Foster et al., 2010; Luo et al., 2011).

Regarding deep convective clouds, some studies have investigated their vertical structure and precipitation characteristics over the Tibetan Plateau (e.g., Fu et al., 2020). Such as, Luo et al. (2011) compared the frequency of occurrence, horizontal and vertical structure of deep convective clouds over the Tibetan Plateau and several typical regions. Fu et al. (2008, 2017) demonstrated that the terrain has a significant impact on the vertical structure of deep convection in the Tibetan Plateau, based on TRMM data. In addition to the vertical structure, previous studies have also investigated the precipitation characteristics of deep convection. Wang et al. (2019) indicate that deep convective precipitation is the primary precipitation type over the



Tibetan Plateau, and the intensity of extreme precipitation will increase significantly with climate warming. In terms of regional differences, the southeastern Tibetan Plateau experiences more convective systems, with 20%-30% of them associated with shear line, vortex, and low-pressure systems, while 70%–80% are formed locally (Hu et al., 2017). Compared to studies on precipitation characteristics, research on the cloud radiative effects (CRE) of deep convective clouds has received less attention over the Tibetan Plateau. Specifically, most studies on the CRE of deep convective clouds only focus on the deep convective cores, rather than the entire deep convective systems. A complete deep convection system (DCS) should include both the deep convective core (DCC) and the anvils. The DCCs are important for the atmospheric hydrological cycle because of the heavy and widespread precipitation, but the non-precipitating anvils cannot be ignored either, as they have distinct radiative effects and more extensive spatial coverage (Feng et al., 2011). Based on surface observations from the SGP site (e.g. total sky imager and condensation particle counter) and satellite retrievals, Yan et al. (2014) even found the opposite effect of aerosol loading on the CRE of DCCs and anvils. They noted that increasing aerosol loading can thicken DCCs, resulting in a cooling effect, while also expanding the extent of anvil clouds, which has a warming effect. Thus, the CRE of entire deep convection systems should depends on the relative contributions from DCCs and anvils. Therefore, a systematic analysis of the radiative effects of whole DCSs over the Tibetan Plateau should be essential to reduce the uncertainties of climate change prediction over the Tibetan Plateau.

In general, both the structure and precipitation of DCSs are influenced by meteorological factors and aerosol loading (Khain et al., 2005; Fan et al., 2009; Zhang et al., 2022; Zhao et al., 2022). For example, Ekman et al. (2007) found, based on model and simulated cases, that the extent of the anvils of the convective cloud is mainly dependent on the updraft velocity. Gong et al. (2020) demonstrated that frozen particle growth and convective system organization are promoted by strong wind shear and high ambient humidity. Aerosols are commonly considered to enhance convective precipitation through indirect aerosol effects (Xiao et al., 2023). Aerosols inhibit warm-rain processes by decreasing particle sizes and creating a narrow droplet spectrum (Squires and Twomey, 1960; Warner and Twomey, 1967; Warner, 1968; Rosenfeld, 1999). The freezing process releases a large amount of latent heating in small droplets, which promotes the development of cumulus clouds into thicker deep convective clouds and more convective precipitation (Andreae et al., 2004; Koren et al., 2005; Rosenfeld et al., 2008; Koren, et al., 2010; Chakraborty et al., 2018; Pan et al., 2021). Continued advances in instrumental technology have

provided strong support for the research of aerosol-cloud-precipitation interactions. Jiang et al. (2018) used the CloudSat/CALIPSO datasets to show that aerosols significantly affect the development of deep convection, with the enhancing or inhibiting effect of aerosols depending on their type and concentration. Based on space-borne and ground-based observations, Sun et al. (2023) found a boomerang-shaped aerosol effect that ranged from promoting to inhibiting the top
height of convective precipitation. They also found that aerosols can have different effects on precipitation rates at different levels. However, the impacts of aerosol and meteorological factors on the structure and precipitation of DCSs are coupled. Fan et al. (2009) found that wind shear influences the effect of aerosols on convection. Under strong wind shear, the increase in aerosol loading will always suppress convection, and under weak wind shear, increased aerosols will invigorate convection. Furthermore, the quantification of direct aerosol invigoration by microphysical effects, also known as primary aerosol
convective invigoration (PAI), has been a challenging issue due to the interference of covarying meteorology-aerosol invigoration (MAI) effects. To address this problem, Zang et al. (2023) utilised an artificial neural network to distinguish the PAI and MAI. Their findings revealed that PAI could explain approximately two-thirds of the observed total aerosol-driven variation in cloud top height. However, these studies mainly focus on the tropics, and it is unclear whether these conclusions can apply to the Tibetan plateau. To address this, the de-coupled effects of aerosols and meteorological factors on the
development and precipitation of DCSs over the Tibetan Plateau and contrastive area will be discussed in this study.

This study systematically explores the differences in the structure, radiative effects, and precipitation of DCSs between Tibetan Plateau and tropical ocean, which was selected as a point of contrast. In particular, this study also discusses the impact of meteorological parameters and aerosol loading on the structure and precipitation of DCSs. The paper is structured as follows: Section 2 outlines the data and methodology. Section 3 presents the results on the structure, radiative effects, and precipitation
of the DCSs in different regions, as well as the influence of meteorological parameters and aerosol loading on them. Finally, Section 4 concludes this study and suggests ideas for further research.



## 2 Data and methods

In this study, the DCSs are identified based on CALIPSO and CloudSat datasets (2B-CLDCLASS-lidar) from 2006 to 2019 over the Tibetan Plateau (TP; 30°N–37°N, 78°E–103°E; Fig. S1) and tropical oceans (TO; 0°–7°N, 68°E–93°E).

Moreover, the radiative effects and precipitation characteristics of DCSs and the influence of meteorology and aerosols are also explored using some auxiliary datasets (e.g., reanalysis datasets). The information of all the datasets and the parameters involved are summarized in Table 1.

| Data Source | Subset | Temporal Coverage | Temporal Resolution | Spatial Resolution | Parameters Selected |
|---|---|---|---|---|---|
| CALIPSO and CloudSat | 2B-CLDCLASS-lidar | 2006-2019 | Orbital Profiles | Orbital Profiles | Cloud Layer Type, Cloud Layer Base Height, Cloud Layer Top Height, Cloud Layer, DEM elevation |
| CALIPSO and CloudSat | 2B-FLXHR-lidar | 2006-2011 | Orbital Profiles | Orbital Profiles | TOACRE, BOACRE, Heating Rate |
| GPM | IMERG | 2006-2019 | Half hour | 0.1°×0.1° | Precipitation_Cal |
| ERA5 | ERA5 hourly data on pressure levels | 2006-2019 | 1 hour | 0.25°×0.25° | u&v component of wind, Relative humidity, Specific humidity, Temperature, Vertical velocity |
| MERRA-2 | M2I3NVAER | 2006-2019 | 3 hours | 0.625°×0.5° | Aerosol mixing ratio |

**Table 1: The temporal coverage and resolution of datasets used in this study.**



## 2.1 Satellite products

The DCSs are identified using 2B-CLDCLASS-lidar, which combines CloudSat's cloud-profiling radar (CPR) and Cloud Aerosol Lidar with Orthogonal Polarization (CALIOP) on board CALIPSO measurements to classify clouds into eight types, including deep convective (cumulonimbus), cumulus etc. This is due to the fact that CPR can penetrate the optically thick cloud layer to explore its internal structure, while lidar is more sensitive to optically thin cloud layers. As a result, the product provides a more complete cloud vertical structure by leveraging the respective advantages of lidar and radar (Wang,

2019). The 2B-CLDCLASS-lidar product is capable of identifying DCSs with both thick DCCs and thin anvils. Regrettably, due to the battery aging of CloudSat (Witkowski et al., 2018), the 2B-CLDCLASS-LIDAR data is only available for the entire day period from 2006 to 2011 and solely during the daytime from 2012 to 2019. Consequently, this study does not analyse the discrepancies between DCSs during the day and night.

       2B-FLXHR-lidar, which inputs the CloudSat/CALIPSO cloud mask and lidar-based aerosol retrievals to a broadband

radiative transfer model in order to provide the radiative fluxes and atmospheric heating rates profiles (Henderson and L'ecuyer, 2020), is used to analyse the CRE and heating rates of DCCs and DCSs in this study. It is important to note that CloudSat will no longer operate during nighttime due to the battery anomaly (Witkowski et al., 2018). For the analysis of radiative effects of DCSs, we have selected the time period from 2006 to 2011, which coincides with the availability period of 2B-FLXHR-lidar. This reduces the sample size, but it is still adequate to obtain statistically significant results to some extent (sample size of

DCSs: TP-116; TO-623). Only daytime data are utilized for the analysis of shortwave CRE and heating rates, while longwave CRE and heating rates are derived from both daytime and nighttime data. To exclude the influence of other cloud systems that overlap with deep convective systems, we have only considered the CRE and heating rates of single-layer cloud profiles. The clouds radiative heating (CRH) is defined as the heating rates difference between the all-sky and clear-sky profiles. The column integral of cloud heating rates is comparable to CRE in the atmosphere (ATMCRE) and $1W/m^2/100$ hPa of ATMCRE

corresponds to about 0.085 K/d of CRH (Yan et al., 2016). Here, it is worth noting that due to the limited sample size of the radiation data, the radiative effects of DCSs are only analysed in terms of their characteristics, and the factors influencing them are not investigated further.



The precipitation of DCSs is derived from the GPM_3IMERGHH product of the Integrated Multi-satellitE Retrievals for GPM (IMERG) Level-3 final run. This product merges precipitation estimates from various precipitation-relevant satellite passive microwave (PMW) sensors comprising the GPM constellation into half-hourly and 0.1°×0.1° (roughly 10×10 km) fields (Huffman et al., 2019). In the precipitation evaluation over Beijiang River Basin of China, the IMERG final run shows high correlation coefficient (0.63) and low relative bias (0.92%) with ground observations (Wang et al., 2017). Due to high spatial and temporal resolution, this precipitation dataset has obvious advantages in accurate matching with DCC samples recognized by 2B-CLDCLASS-lidar. It provides us with confidence to analyse the relationship between precipitation features of DCCs and meteorology. In addition, this study only analyses precipitation grids (precipitation > 0) that are spatially closest to DCC profiles and temporally closest to the detected time of each DCC. The mean precipitation is obtained by averaging these grids to match each DCC.

## 2.2 Reanalysis datasets

ERA5 is the fifth generation of ECMWF reanalysis for global climate and weather data from the past eight decades. The data is available from 1940 onwards. Reanalysis combines model data with observations from around the world using the laws of physics to create a globally complete and consistent dataset. It optimally combines previous forecasts with the most recent observations to produce new best estimates of atmospheric conditions (Hersbach et al., 2020). Compared to ERA-Interim, ERA5 offers superior spatial and temporal resolution, as well as improved consistency between forecast models and observations (Hoffmann et al., 2019). ERA5 reanalysis is frequently employed in cloud and precipitation research, particularly in areas where ground-based observations are unavailable, such as the TP (Chen et al., 2020). In this study, the ERA5 hourly data on 37 pressure levels from 1000 hPa to 1 hPa with resolution of 0.25°×0.25° are used to provide the meteorological fields of DCSs (e.g. u&v component of wind, relative humidity, specific humidity, temperature, and vertical velocity). The calculation methods of meteorological factors are detailed in Sect. 2.4.2, while the relationship between meteorological factors and precipitation characteristics of DCSs is discussed in Sect. 3.3.

The 3-hourly aerosol reanalysis datasets (M2I3NVAER) derived from the Modern-Era Retrospective analysis for Research and Applications Version 2 (MERRA-2) with a spatial resolution of 0.625°×0.5° (longitude×latitude) are used to



demonstrate the impact of aerosol loading on DCSs. MERRA-2 is the latest atmospheric reanalysis of the modern satellite era produced by NASA's Global Modeling and Assimilation Office (GMAO). It combines recent measurements of atmospheric states and remotely sensed aerosol optical depths to provide the aerosol reanalysis (Buchard et al., 2015; Molod et al., 2015;

Randles et al., 2016; Gelaro et al., 2017). It has been widely evaluated and used to study the interaction between aerosol and cloud (Douglas and L'ecuyer, 2020). M2I3NVAER provides assimilations of aerosol mixing ratio parameters at 72 model layers from 985 hPa to 0.01 hPa. The parameters include dust, sulphur dioxide, sulphate aerosol, sea salt, hydrophilic black carbon, and hydrophilic organic carbon. Fine aerosols with a radius of less than 1 μm are the best proxy for cloud condensation nuclei (CCN) from MERRA-2 (Pan et al., 2021). They are also the best proxy for describing the aerosol invigoration on deep

convection when compared to aerosol optical depth and coarse aerosols (Pan et al., 2021). The study uses the mass concentrations of fine aerosols with a radius < 1 μm under cloud base as a proxy for aerosol loading. The mass concentration of fine aerosols at all grids where each DCC is located is averaged to match each DCS, including the corresponding DCC. It is important to note that the aerosol information is selected from the first hour without precipitation before the DCSs are detected on the same day, as wet deposition processes can influence the results. The aim is to clarify the role of aerosols in the

development and precipitation of DCSs.

**2.3 Method**

**2.3.1 Identification of DCSs**

DCS comprises of two components: DCC and anvil clouds that are connected to it. It is important to note that, unlike previous studies that focused on multiple profiles identified as DCS/DCC (Peng et al., 2014; Zhang et al., 2022), in this study,

one DCS/DCC refers to the collection of spatially continuous cloud profiles. Here, DCCs are defined as adjacent profiles that contain cloud layers classified as 'deep convective' type in 2B-CLDCLASS-lidar. To ensure accuracy, only samples that meet the following two criteria, as per the methods of Luo et al. (2011), are retained, thereby excluding the influence of broken deep convection or cumulus on the results: The criteria for identifying deep convective clouds (DCC) are: 1) the distance between the lowest cloud base of DCC and the ground or sea surface is less than 3 km, and 2) the maximum cloud top height of DCC

is more than 12 km above the mean sea level.



Once the DCCs have been identified using the methods described above, the DCSs containing the corresponding DCCs can be identified based on the cloud top and base heights of the adjacent profiles. To achieve this, each vertical profile is divided into height bins at 240m intervals, which corresponds to the vertical resolution of the CPR. The cloud top and base heights from 2B-CLDCLASS-lidar are then used to determine whether each bin is cloudy. The binary image is composed of adjacent profiles with cloudy pixels (assigned to 1) and clear pixels (assigned to 0). The 'bwboundaries' function (Gonzalez et al., 2004) is then used to trace the boundaries of the binary image and separate and identify disconnected cloud objects. The cloud objects containing the DCCs picked out in the previous step are defined as the DCSs in this study. The anvil in this paper is defined as the part of the DCSs other than DCCs.

Fig. 1 shows the example of DCS and flow chart for identifying DCCs and DCSs. The width of the DCSs/DCCs is calculated by multiplying the number of profiles by 1.1 km (horizontal spacing of adjacent profiles), and the thickness is calculated as the difference between the maximum cloud top height and the minimum cloud base height of the DCCs.

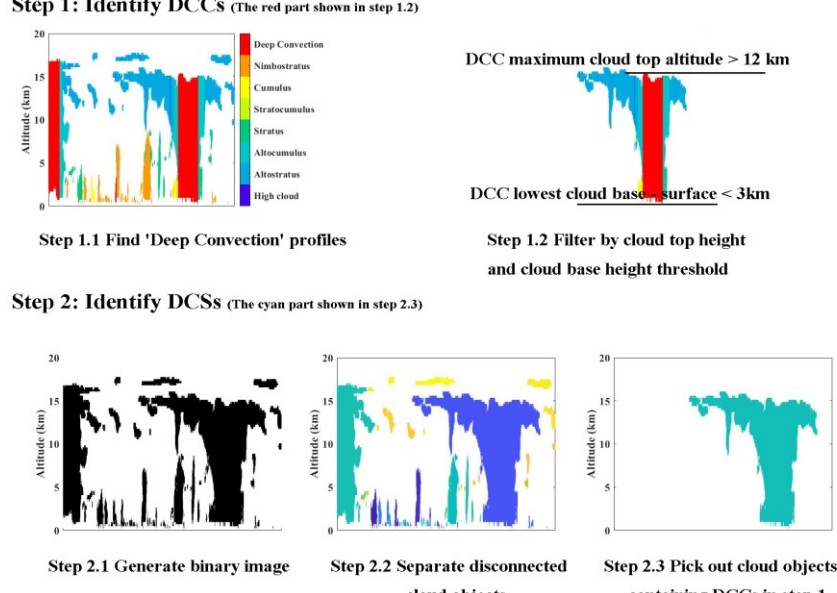

**Figure 1: Flow chart for DCCs and DCSs identification.**



### 2.3.2 Calculation of meteorological factors

Previous studies have pointed out that the wind shear plays a significant role in developing of DCSs (Fan et al., 2009). In

this study, we investigate the relationship between the u and v components of wind, vertical velocity and the development of

DCSs using hourly ERA5 data on pressure levels with a horizontal resolution of 0.25° × 0.25° (Hersbach et al., 2019).

Following the methodology of Di Giuseppe and Tompkins (2015) and Li et al. (2018), the u and v components of the wind are

projected onto the CloudSat overpass track to obtain the scene-averaged, along-track horizontal wind $\mathcal{V}$. The wind shear

($d\mathcal{V}/dz_{i,j}$) between any two atmospheric layers $i$ and $j$ may be expressed as:

$$dV/dz_{i,j} = \frac{max\{\mathcal{V}_i; \mathcal{V}_j\} - min\{\mathcal{V}_i; \mathcal{V}_j\}}{\mathcal{D}_{i,j}} \tag{1}$$

where $\mathcal{V}_i$ and $\mathcal{V}_j$ are the horizontal winds at layers $i$ and $j$, respectively, and the $\mathcal{D}_{i,j}$ is the layer separation distance.

The layers $i$ and $j$ are determined by the maximum and minimum heights where the first profile and the last profile of DCSs

are located.

The impact of conditional instability of the moist convection on cloud spatial scale has been demonstrated (Mieslinger et

al., 2019). Following Li et al.'s (2018) approach, we compute the vertical gradient of the saturated equivalent potential

temperature (∂θ_es/∂z) between the layers $i$ and $j$, as described in the wind shear calculations.

$$\begin{cases} \theta_{es} = \theta exp\left(\frac{L_v r_s}{C_p T}\right) \\ \theta = T\left(\frac{1000}{p}\right)^{0.286} \\ L_v = 2.5\times10^6 - 2323\times(T-273.16) \\ r_s = \frac{sh}{RH\times(1-sh)} \end{cases} \tag{2}$$

where $\theta$ is the potential temperature, $L_v$ is the latent heat of vaporization, $r_s$ is the saturation mixing ratio, $C_p$ is the

specific heat capacity at a constant pressure, and $T$ is the atmospheric temperature. The larger ∂θ_es/∂z, the more stable the

atmosphere.

Previous studies have indicated that vertical velocity plays a crucial role in the development and precipitation of DCSs

(Khain et al., 2005; Xiao et al., 2023). To analyse the impact of high-level vertical motion on DCSs development and

precipitation, we extracted the vertical velocity at 500 hPa in the TO region from ERA5, as suggested by Li et al. (2017a) and



Zhang et al. (2022). The study by Zhang et al. (2020) suggests using the vertical velocity at 250 hPa as a proxy for high-level

vertical motion in the TP region due to its high-altitude surface and distribution of vertical winds.

Wind shear and $\partial\theta_{es}/\partial z$, which are linked to the horizontal development of DCSs, are averaged at all grids where each

DCS is located. The vertical velocity is averaged at the grid points where the DCC is located. Meteorological factors in the

hour before the DCS is detected are used for correlation analyses. Additionally, the movement of DCSs/DCCs under advection

is ignored. This study discusses the relationship between meteorological factors and aerosol-cloud-precipitation interactions.

However, it is important to note that the apparent correlations do not necessarily imply causality.

### 2.3.3 Calculation of tropopause height

Overshooting convection significantly impacts stratospheric water vapour (Jensen et al., 2007). To analyse the proportion

of DCSs that penetrate the tropopause in different regions, we first need to calculate the tropopause height by using the

temperature profiles from ERA5 hourly reanalysis data on pressure levels (Hersbach et al., 2020). The data has been regridded

to a regular lat-lon grid of 0.25 degrees on 37 vertical levels. The high spatiotemporal resolution data enables obtaining

temperature profiles that match the location of the DCCs at the same hour as the convection samples are observed. The

reliability of ERA5 data has been verified (Xian and Homeyer, 2019) and it has been confirmed that accurate tropopause

heights can be obtained based on ERA5 (Sun et al., 2021). Using the ERA5 temperature profiles that match DCSs (closest to

the location and time of DCSs being observed by satellite), we compute the vertical lapse rate profile according to the methods

in Reichler et al. (2003). The profile is then interpolated on a 100 m vertical grid. The tropopause height is calculated based

on the definition provided by the World Meteorological Organization (WMO, 1957): "(1) The first tropopause is defined as

the lowest level at which the lapse rate decreases to 2 K km$^{-1}$ or less, provided also the average lapse rate between this level

and all higher levels within 2 km does not exceed 2 K km$^{-1}$. (2) If there exists a layer above the first tropopause at which the

lapse rate exceeds 3 K km$^{-1}$ in any altitude range up to 1 km above it, a second tropopause is defined according to the criterion

in (1). This second tropopause may be within or above that 1 km layer."



# 3 Results

## 3.1 The vertical structure of DCSs

This study analyses the characteristics of DCSs by focusing on whole cloud clusters, rather than individual profiles or

grid boxes as in previous studies (Feng et al., 2011; Peng et al., 2016). Each cloud cluster identified as a DCS is considered as

one sample. Table 2 shows that there are more DCSs in the TO (977) than in the TP (240), due to differences in moisture and

thermal conditions. Peng et al. (2014) found that there are more DCSs located at low latitudes than mid-latitudes, which is

consistent with our results. The TO is located in the rising branch of the Hadley cell, the large-scale circulation also provides

favourable conditions for convection (Bowen, 2011). It should be noted that the TP region is exclusively defined for the central

Tibetan Plateau, aiming to minimize the influence of surface gradients on the formation of deep convective clouds (DCSs).

Thus, this definition excludes the southern slope of Tibetan Plateau where convection occurs frequently. Compared to the TP

region, the DCSs over the tropical ocean are wider and thicker. The average width and thickness of DCSs over the TO are

637.1 km and 14.5 km, respectively. The average horizontal scale of DCSs over the TP region (about 258.8 km) is less than

half of those over the TO. Meanwhile, the DCSs over the TP are thinner, measuring 9.8 km. Table 2 and Fig. 2 show that both

the convective core and anvil cloud are narrower over the TP. And the proportion of DCC in DCS is greater in the TP region

at 17.2 %. According to Fig. 2, the width of DCCs and DCSs in the TP is mostly concentrated at 10 km and 100 km, respectively,

while TO exhibits greater inter-sample variation in width with a wider range. The differences in deep convective features

between the TP and the TO are closely linked to their unique local environment and terrain. The TP region is influenced by

westerlies and the Indian monsoon, as well as local recycling, which involves evaporation, convection, and droplet re-

evaporation (Yao et al., 2013). The lower height of the level of neutral buoyancy, convective available potential energy, and

total precipitable water in the TP restrict the development of DCSs (Luo et al., 2011). Furthermore, the moisture supply in the

TP is limited by topography, resulting in a compressibility effect that reduces the cloud thickness of DCSs. Compared with

TO, the ratio of DCCs/DCSs penetrating tropopause is less over TP and around 18.3%/28.8%. Even so, previous studies have

found that deep convective processes on the TP contribute more to stratospheric pollutants than those in the entire tropical

region (Randel et al., 2010). Therefore, it is important to pay particular attention to overshooting convection on the TP. In the



tropical regions, numerous studies have shown that DCSs are most commonly observed over this region (Sassen et al., 2009; Savtchenko, 2009). The main factor contributing to the generation of DCSs is convective instability caused by radiative heating (Peng et al., 2014). Despite the challenges posed by the higher tropopause height at lower latitudes, up to 38.1% of DCSs in the TO region can penetrate the tropopause (as shown in Table 2). Furthermore, studies have indicated that if the tropical

tropopause layer is initially supersaturated with respect to ice, the DCSs that penetrate the tropopause will reduce the humidity. This is due to the excess vapour condensing on the ice crystals (Jensen et al., 2007).

| Region | Sample number | Width of DCSs (km) / SD | Width of DCCs (km) / SD | Width of anvil (km) / SD |
|---|---|---|---|---|
| TP (total) | 240 | 258.8/310.9 | 24.0/20.6 | 234.8/306.9 |
| TO | 977 | 637.1/563.4 | 61.0/67.3 | 576.1/541.5 |
| TP (NW) | 18 | 230.5/178.8 | 18.3/16.6 | 212.2/169.6 |
| TP (NE) | 51 | 205.3/196.7 | 25.9/21.0 | 179.4/187.5 |
| TP (SW) | 79 | 229.4/242.4 | 22.1/15.8 | 207.3/237.1 |
| TP (SE) | 92 | 319.3/412.3 | 25.8/24.3 | 293.6/410.6 |

| Region | DCCs /DCSs[a] (%) | Thickness of DCCs (km) / SD | DCCs/DCSs penetrating tropopause (%) | Mean precipitation of DCCs (mm hr$^{-1}$) |
|---|---|---|---|---|
| TP (total) | 17.2 | 9.8/1.5 | 18.3/28.8 | 1.2 |
| TO | 15.6 | 14.5/1.4 | 26.8/38.1 | 3.5 |
| TP (NW) | 12.4 | 9.3/1.4 | 22.2/27.8 | 0.8 |
| TP (NE) | 19.1 | 10.0/1.5 | 23.5/31.4 | 1.7 |
| TP (SW) | 16.5 | 9.7/1.5 | 17.7/27.9 | 0.9 |
| TP (SE) | 17.8 | 9.8/1.5 | 15.2/28.3 | 1.3 |

[a]The average of the ratio of the width of DCC in each DCS

**Table 2: The spatial statistics of DCSs in different subregions. The definition of different parts of TP are as follows: TP(NW) (33.5°N–37°N, 78°E–90.5°E); TP(NE) (33.5°N–37°N, 90.5°E–103°E); TP(SW) (30°N–33.5°N, 78°E–90.5°E); TP(SE) (30°N–33.5°N, 90.5°E–**

**103°E). SD is an abbreviation for standard deviation.**





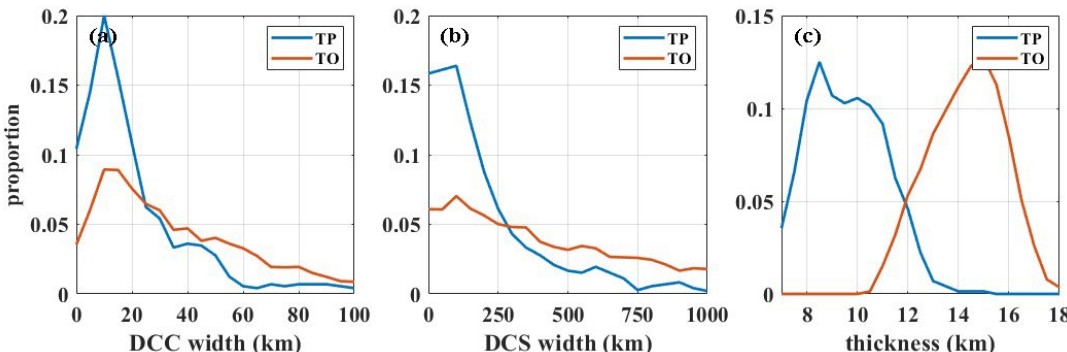

**Figure 2: The proportion of each bin of the width of DCCs (a), the width of DCSs (b), and the thickness of DCCs (c) in the total sample of TP, and TO.**

To investigate the internal differences of the TP further, we divided it into four subregions using the coordinates 33.5°N

and 90.5°E. According to Table 2, DCSs are mainly observed in the southern TP, which accounts for 71% of the total samples. This is due to the more significant influence of the Indian monsoon on the southern TP, which releases large amounts of latent heat and abundant moisture during summer (Jiang et al., 2016; Zhang et al., 2017). The southeastern TP had the largest number of samples (92) and widest DCSs (319.3 km) with widest anvil (293.6 km). The southwestern TP had the second most DCSs (79), likely due to up-and-over convective storms and upslope moisture transport from the Indian subcontinent (Dong et al.,

2016). In contrast, fewer DCSs form in the northwestern TP, and they are also thinner, with narrower anvil and convective core. Warm and moist air from the Indian Ocean and nearby areas can promote convection over the southern TP, but it is difficult to transfer to the northern TP (Xu et al., 2003). The evidence of lower specific humidity and vertical velocity in the northern TP (Zhang et al., 2020) also shows the difficulty in the development of DCSs.

Fig. 3 shows the seasonal sample number, width, and thickness of DCSs in different seasons. It is worth noting that there

are too few samples in spring to be representative, and therefore these DCSs will not be analysed. The TP region experiences significant seasonal differences in cloud development, which is affected by the Indian monsoon (Yao et al., 2012). In summer, the TP continuously attracts moist air from the low-latitude ocean (Xu et al., 2008), acting as a strong "dynamic pump" (Wu and Zhang, 1998). The lower branch of atmospheric flows packed with these moisture (Bai and Xu, 2004) rises along the south side of the plateau, and causes frequent convections and precipitations (Xu et al., 2003). As shown in Fig. 3a, b, and c, the

DCSs are most frequent, widest, and thickest during the summer. The average width and thickness of DCSs over TP region





during summer season are around 266 km and 9.9 km, respectively. Additionally, DCS development is significantly more dominant in the summer than in other seasons for each subregion of the TP region (see Fig. 3d, e, f). Fig. 3e shows that the DCSs is most vigorous developed in the southeastern TP. This phenomenon is due to the abundance of water vapour (Zhou et al., 2012) and the favourable dynamic and thermal conditions for convection in the southeastern TP in summer. Under the

control of a high-pressure system (Zhang et al., 1997), however, there is no deep convective system (DCS) over the TP during winter. In comparison to the TP, which is impacted by monsoons, the thickness of DCSs over the TO does not vary much by season. Summer convection in the tropical oceans is the least abundant (Fig. 3a) but the most horizontally extensive, with an average width of 811 km (Fig. 3b). The thickness of DCSs remains consistent throughout the seasons, at approximately 14.5 km (Fig. 3c).

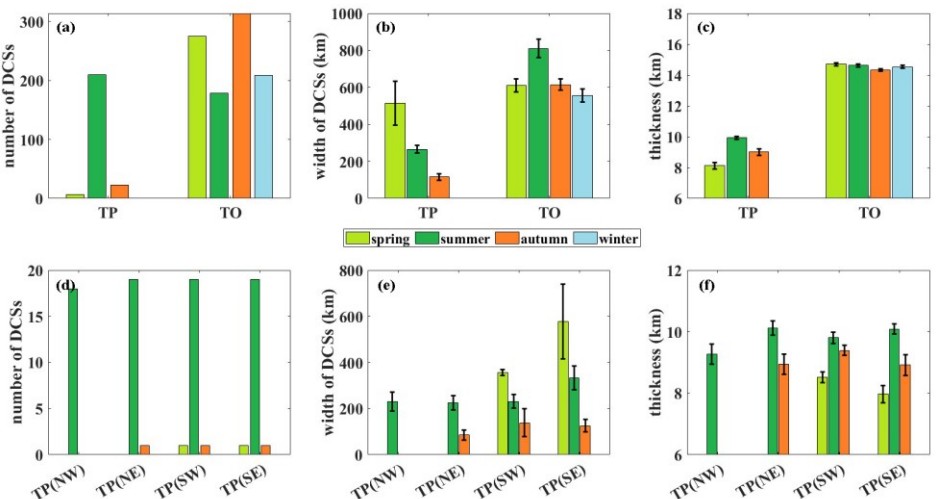


**Figure 3: The sample number (a, d), the width (km) (b, e) and the thickness (km) (c, f) of DCSs in different regions in different seasons (in the boreal hemisphere). The error bars represent the standard error for the sample mean (SEM, SEM= standard deviation/$\sqrt{n}$, n is sample number).**

**3.2 Cloud radiative effects of DCSs**

DCSs have a significant impact on the radiative flux in and out of the earth-atmosphere system. Additionally, deep convection plays a crucial role in the atmosphere's interior. For instance, convective cloud top radiative cooling can promote cirrus formation in the tropopause transition layer (Sassen et al., 2009). DCSs are not comprised of a single uniform cloud





type. Therefore, anvils and convective cores with different cloud characteristics have varying cloud radiative effects (Feng et al., 2011; Hartmann et al., 2001). For example, the outgoing longwave irradiances are higher for anvil clouds than for

convective cores, while the reflected shortwave flux is greater for convective cores (Feng et al., 2011). Table 3 displays the average net CRE of DCSs, DCCs, and anvils from different regions. The probability density distribution of net CRE for different regions is also shown in Fig. 4. Table S1 present the shortwave (SW) and longwave (LW) CREs, respectively. To exclude the effects of other cloud systems, only the CRE of single-layer cloud profiles in DCSs is considered in this section. As the radiation data are only available for nearly five years, the sample size is reduced compared to Sect. 3.1. More details

are provided in Sect. 2.2. Table 3 shows that the DCCs have the largest net CRE absolute value, while the net CRE absolute value of anvils is the smallest, and that of DCSs is the second largest. This indicates that the anvils, with thin optical thickness but high cloud height, partly offsets the strong cooling effect caused by optically thick DCCs in the whole DCSs. Anvil clouds have a limited impact on the reduction of incoming SW radiation, but they can emit LW radiation to some extent (as shown in Table S1). These findings are consistent with the previous study by Feng et al. (2011). Table 3 shows differences between

regions, with the DCCs in TP having the weaker radiative cooling effect, likely due to their thinness compared to other regions. The average net top-of-atmosphere (TOA) and bottom-of-atmosphere (BOA) CRE are -407.3 W m$^{-2}$ and -569.7 W m$^{-2}$, respectively. This results in an average radiative heating of the inner atmosphere of 162.4 W m$^{-2}$. In contrast, DCCs in the TO have the greatest radiative cooling effect with a net TOA CRE of -526.5 W m$^{-2}$ and a net BOA CRE of -703.9 W m$^{-2}$, resulting in the CRE in the atmosphere (ATM) being 177.4 W m$^{-2}$. Table S1 shows that the regional differences in DCCs TOA CRE are

mainly due to differences in SW CRE. In particular, the LW BOA CRE of DCCs in the TP is significantly greater than that in the TO. This phenomenon was also observed by Yan et al. (2016), who proposed that it may be due to the uplifted topography and associated unique vertical profiles of temperature and water vapour. To identify the cause of the significant LW BOA CRE in the TP, we examined the clear-sky LW flux and the cloudy LW flux at the BOA. The LW CRE was calculated by subtracting the net clear-sky LW flux from the net LW cloudy flux. The findings, as shown in Fig. S2 and S3, indicate that the TP varies

from the TO primarily in terms of downward clear-sky LW flux. The lower humidity atmosphere of the TP is less capable of capturing LW radiation, resulting in a particularly small clear-sky downward LW flux (Yang et al., 2010). This contributes to the more negative net (downward minus upward) clear-sky LW flux in the TP. Although the colder cloud base temperatures



at the TP (as shown in Fig. S9) result in a smaller net cloudy LW flux, it does not change the fact that the DCCs in the TP have a larger LW BOA CRE. Also due to the more negative clear-sky LW flux, the DCSs and anvils in the TP have larger LW BOA

CRE than those in the TO (Table S1). The unique topography and water vapour conditions of the TP may make the effect of clouds on radiation more efficient. As explained earlier in this sub-section, the difference in CRE between DCCs and DCSs is due to the effect of anvils. At the TP, where the anvils are narrower, the net TOA CRE of DCSs is -307.2 W m$^{-2}$, which is 100.1 W m$^{-2}$ higher than the CRE of DCCs. In contrast, the absolute value of the TOA CRE of DCSs is smaller in the TO region than in the TP, despite the fact that the radiative cooling of DCCs at TOA is stronger in the TO. The TO region has

more massively developed DCSs, which also have wider anvils. Thus, in the TO region, the anvils with larger (absolute minimum) TOA CRE (around -157.4 W m$^{-2}$) has the more obvious offsetting effect on the NCRE of DCCs. In contrast, in the TP region, where the development of DCSs is relatively less pronounced, the TOA CRE of the anvils is -249.0 W m$^{-2}$. It should be noted that the relatively thinner anvils in the TP (averaging 4.3 km) results in more radiative cooling compared to the anvils in the TO (averaging 5.5 km). The disparity in TOA CRE between the anvil clouds in the two regions is mainly due to the fact

that the anvils in the TP has more negative SW CRE (Table S1). Fig. S4 shows that in cases with comparable thickness, the SW TOA CRE of anvils in the TP is more negative than that observed in the TO. Luo et al., (2011) found that the DCCs in the TP have more densely packed cloud tops dominated by larger particles, as indicated by the smaller distance between cloud top height and radar echo-top (10 dBZ) height. This is a common disparity between continental and oceanic deep convection (Liu et al., 2007). It is also plausible that this dissimilarity extends to the anvils of DCSs.

| Region | TOA NCRE (W m$^{-2}$) | | | BOA NCRE (W m$^{-2}$) | | | ATM NCRE (W m$^{-2}$) | | |
|---|---|---|---|---|---|---|---|---|---|
| | DCC | DCS | anvil | DCC | DCS | anvil | DCC | DCS | anvil |
| TP | -407.3 | -307.2 | -249.0 | -569.7 | -419.3 | -335.5 | 162.4 | 112.1 | 86.5 |
| TO | -526.5 | -240.8 | -157.4 | -703.9 | -356.7 | -253.1 | 177.4 | 115.9 | 95.7 |

**Table 3: The mean net cloud radiative effects (NCRE) of DCSs, DCCs and anvil in different regions.**



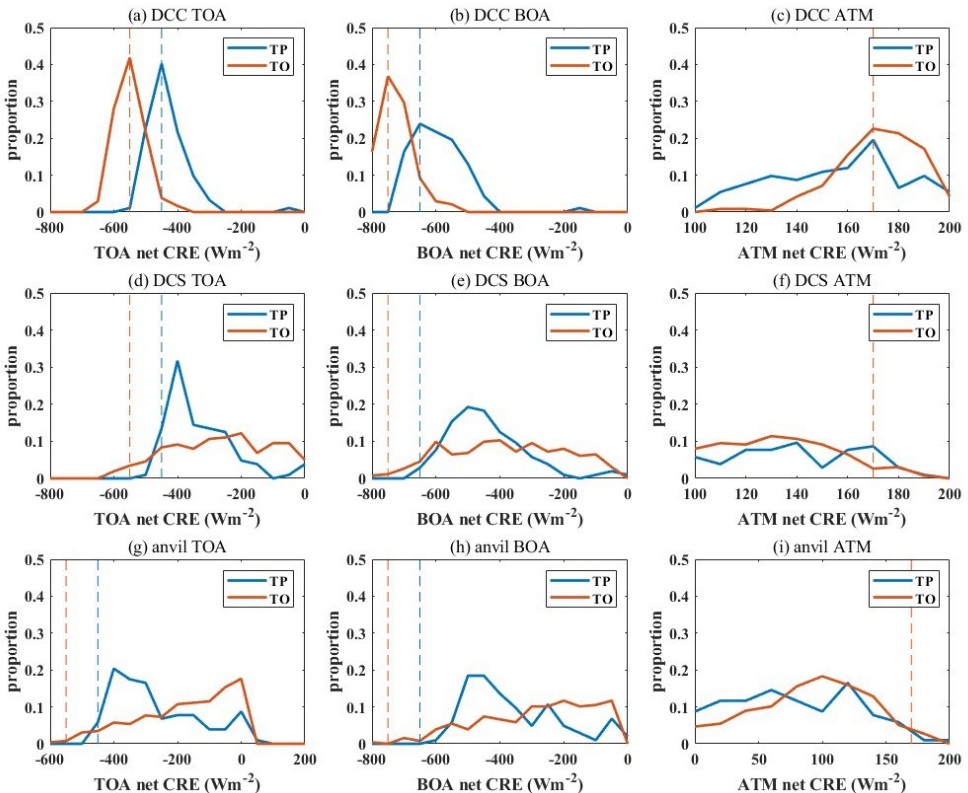

**Figure 4: The proportion of each bin in the total sample of DCC, DCS and anvil in the TP (blue), TO (red) of the net CRE at the top of the atmosphere (TOA) (a, d, g), at the bottom of the atmosphere (BOA) (b, e, h) and in the atmosphere (ATM) (c, f, i). The dash lines show the peak of DCC.**

The sample distribution of CRE for DCSs in different regions is illustrated by the proportion of numbers in each bin to the total sample size (refer to Fig. 4). As illustrated in Fig. 4a, the net TOA CRE of DCCs in the TP is concentrated around -450 W m$^{-2}$. However, 95% of the DCCs in TO have a net TOA CRE less than -450 W m$^{-2}$, resulting in a larger radiative cooling effect. As shown in Fig. S5, the SW CRE of DCCs in the TO is much smaller than that in the TP. The reason for this result is that the DCSs in the tropics are thicker and reflect more SW radiation back to TOA. Additionally, the LW CRE of the

DCCs in the TO is slightly larger than that in the TP. It was found that the TOA in the tropics receives more LW radiation under clear-sky conditions, while DCCs with higher cloud tops and lower temperatures emit less LW radiation. The larger difference between them results in a greater LW CRE for DCCs in the tropics. The concentration of DCCs TOA SW and LW CRE is more pronounced in the TO than in the TP. This is due to the homogeneous subsurface of the TO region and the



thickness of the DCSs, which do not exhibit significant seasonal variations. Similarly, for net BOA CRE in Fig. 4b, DCCs in the TO also exhibit significantly greater radiative cooling, with the net CRE concentrated around -750 W m$^{-2}$, with an absolute value 100 W m$^{-2}$ greater than that in the TP. Fig. 4c shows that both TP and TO's DCCs have a concentration of ATM net CRE at 170 W m$^{-2}$. However, the CRE of DCCs in the TO is larger overall, while that in the TP has a more spread-out distribution. From Fig. S5 (i, and j), it can be seen that the DCCs in the TO have less ATM SW CRE due to stronger reflection of solar radiation with larger thickness. Additionally, the ATM LW CRE of DCCs in the TO is larger due to the higher cloud temperature with lower cloud base. Fig. 4d, e, and f display the net CRE of DCSs in various regions, which is more dispersed than that of the DCCs. One possible reason for this is that the structure of the different DCSs varies considerably, particularly in the TO region, which is partially visible in Fig. 2b. The dashed lines indicate the value at which the CRE of DCCs is primarily concentrated. It is evident that the TOA and BOA CRE of DCSs are generally larger than those of DCCs in both TP and TO. The ATM CRE of DCSs varies greatly between samples in both the TP and the TO, and is much lower than that of the DCCs. Fig. S5 (k, and l) shows that the ATM SW and LW CRE of DCSs in both TP and TO are smaller after adding the anvils into count. The differences between the CRE of DCCs and DCSs are due to the dilution of the thinner and higher anvils, which reflects solar radiation less and emits less LW radiation.

To analyse the ATM CRE further, we calculate the vertical cloud radiative heating rate (CRH) profiles for the DCSs/DCCs. CRH is defined as the difference between the all-sky radiative heating rates and the clear-sky radiative heating rates. Radiative heating is an important component of the Earth system's energy budget (Stephens et al., 2012), and clouds play a crucial role in regulating radiative processes (Slingo and Slingo, 1988, 1991). Passive sensors struggle to identify vertical boundaries of clouds, resulting in a poorly resolved vertical profile of global radiative heating (Mace, 2010; Haynes et al., 2011). However, active sensors such as the CPR onboard CloudSat and CALIOP onboard CALIPSO can provide a more accurate vertical gradient of CRH (Henderson et al., 2013) and a near-global view of the tops and bases of most radiatively active clouds. Currently, vertical CRH profiles are commonly used to analyse the cloud radiative effects on the atmosphere (Lv et al., 2015; Yan et al., 2016; Pan et al., 2020; Zhao et al., 2024). In fact, the CRH (in K d$^{-1}$) is equivalent to the ATM CRE per unit mass (in W/m$^2$/100 hPa) (Yan et al., 2016). Fig. 5 displays the averaged vertical profiles of CRH in various regions. As only sufficient deep convection was identified (cloud top heights above 12km above sea level and cloud base heights within 3km





of the ground; see Sect. 2.4.1 for identification methods), the cloud top height of DCSs in TP and TO did not differ significantly,

and the patterns of CRHs in different regions were generally consistent. Fig. 5a and 5d demonstrate that the negative LW CRH

peak is at 14km for both TP and TO, with similar DCC LW CRH values of -6.0 K d$^{-1}$ and -5.7 K d$^{-1}$, respectively. The radiative

flux exchange between the atmosphere above the cloud and the cloud top causes the LW CRH near the cloud top to be negative.

For DCCs, the LW CRH is positive within 10km. This is due to enhanced infrared cloud radiative warming by upper-level

cloud tops, which prevent LW radiation from escaping to space while radiating towards the surface (Haynes et al., 2013). Near

the surface, there is stronger radiative heating, which is due to the radiative heating of the cloud base. This peak of LW CRH

near the surface is more distinct in the TP, as the colder boundary layer in the TP (Haynes et al., 2013). Note that the TP has a

significant topographical gradient, resulting in fewer samples from height bins below 5.5km and a larger standard deviation.

To ensure the representativeness of the results, height bins with only several valid data are not shown in Fig. 5. The high value

of CRH, the difference between all-sky and clear-sky heating rates, can be analysed separately under all-sky and clear-sky

conditions. Under all-sky conditions, the LW heating rate of the boundary layer is significantly greater in the TP than in the

TO (refer to Fig. S6b, d). This is due to the fact that the cold boundary in the TP can only emit a small amount of upwelling

LW radiation, which is less than the infrared radiation from the cloud above, resulting in a large peak of the all-sky heating

rates. In clear-sky conditions (refer to Fig. S7b, d), the LW cooling primarily results from the emission of water vapour, which

occurs throughout much of the troposphere and decrease above 10km in both regions mainly due to the decrease in saturation

water vapor pressure with decreasing temperature (Hartmann et al., 2001). The less LW cooling in the low-level atmosphere

of the TP is likely attributed to the lower surface temperature and water vapor in the TP (McFarlane et al., 2007). Combined

with the large LW heating under all-sky conditions and small LW cooling under clear-sky conditions, the difference between

the two conditions, LW CRH, appears to be large near the surface of the TP. The DCSs have a smaller average thickness and

lower cloud top, resulting in less LW radiative cooling near the cloud tops and less LW radiative heating in the low level.

There are two peaks of DCSs LW CRH in the TP at 8 km and 14 km, respectively. In the TO, DCSs LW CRH only peaks at

14 km. The distribution of DCSs LW CRH is associated with the difference in cloud tops between the two regions. We checked

the sample distribution of cloud top heights for DCSs in both regions and found the same peaks.



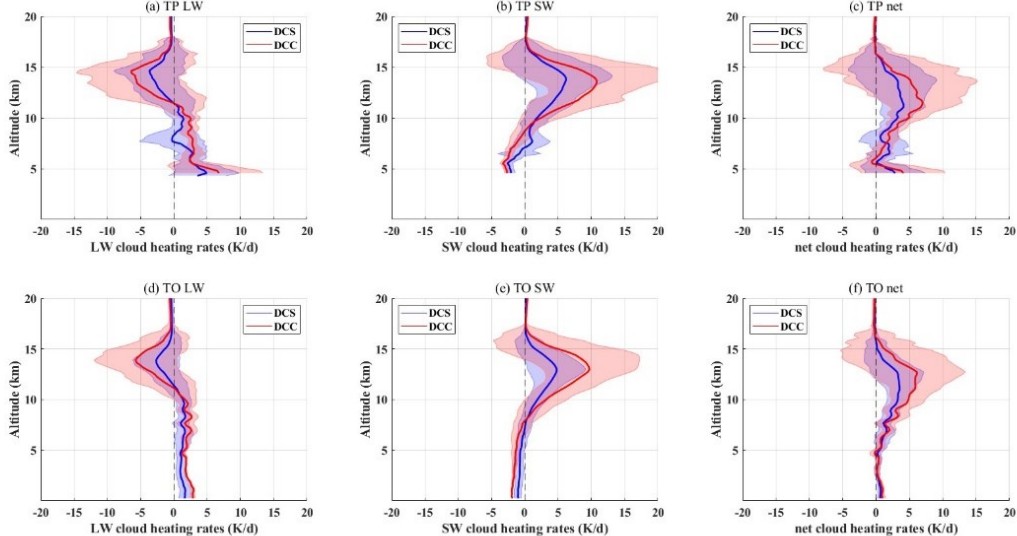

**Figure 5: The mean vertical profiles of longwave (a, d), shortwave (b, e), and net (c, f) cloud radiative heating rates (CRH) (K d⁻¹) in the TP and TO. The red lines represent the results of DCCs and the blue lines represent the results of DCSs. The shadow range represents the standard deviation.**

As shown in Fig. 5 (b, e), the peak of SW CRH is also near the cloud tops at 13 km of 10 K d⁻¹ of DCCs. The cloud tops absorb SW radiation, and tend to reduce SW absorption by the underlying atmosphere by reflection (Haynes et al., 2013; Pan et al., 2018), resulting in maximum SW CRH in the upper troposphere and small SW CRH in the low-level. The DCCs in TP and TO show similar SW heating rates for all-sky conditions, except that the peak in the TP is slightly larger (Fig. S6). The clear-sky SW heating rates are larger in the TP despite having less water vapour and oxygen to absorb SW radiation. The reason for this is that vast majority of TP's DCSs are observed in summer, when the solar zenith angle is low, with more opportunity to absorb SW radiation. But there is no significant seasonal difference in the number of DCSs in the TO compared with the TP. Also, due to this reason, the SW CRH is smaller at lower levels. Similar to the results for LW CRH, the absolute value of SW CRH for DCSs is smaller than that for DCCs when averaged with thinner anvils. Additionally, the second peak of SW CRH for DCSs in the TP is related to the second peak of frequency of the DCSs' cloud top heights.

Combining LW and SW CRH, the net CRH exhibits significant radiative heating in the upper troposphere (Fig. 5 c, f). This heating is more pronounced for DCCs than DCSs. The radiative heating caused by DCSs has great influence on large-



scale circulation variations and the column-integrated radiative heating of upper-level clouds in the tropics accounts for

approximately 20% of the latent heating (Li et al., 2013). Fueglistaler and Fu (2006) proposed that the spatial gradients in

radiative heating rates, caused by the seasonal and spatial differences of tropical convection and associated high cloud

coverage, may be partially responsible for stratospheric mixing. Since the development of DCSs is forced by the high terrain

of the TP (Fu et al. 2008, 2017), both the LW heating layer and the SW cooling layer are shallower for TP DCSs. The

maximum net CRH of DCCs in the TP is 7 K d$^{-1}$ at 11.3 km, and in the TO it is 6 K d$^{-1}$ at 12.7 km. The net CRH of the DCCs

is larger in the TP than in the TO in the low-level atmosphere due to the large LW radiative heating of the boundary layer by

DCCs. The CRH of DCSs has similar features to those of DCCs, but the regional differences are smaller. The differences

between DCSs and DCCs CRH are primarily attributed to anvil clouds, making them predominantly evident in the upper

atmosphere.

**3.3 The effect of meteorological factors and aerosols on the development and precipitation of DCSs**

**3.3.1 The effect of aerosols and meteorological factors on the development of DCSs**

Previous studies have demonstrated that various meteorological factors significantly impact the development of DCSs.

For example, using a three-dimensional numerical cloud model, Weisman and Klemp (1982) find that for a given amount of

buoyancy, higher wind shear leads to more conducive supercell formation. Atmospheric instability, which affects the

entrainment drying and warming, has been found to be associated with cloud size in the observations (Mieslinger et al., 2019).

In addition, previous studies have found that vertical velocity is a significant factor in the development of deep convection

(Khain et al., 2005; Koren et al., 2010; Jiang et al., 2018). In addition to meteorological factors, aerosols can also have a critical

impact on clouds. Aerosols can modify the microphysical and optical properties of DCSs by acting as CCN and ice nuclei (IN)

in the atmosphere (Peng et al., 2016; Li et al., 2017b; Pan et al., 2021; Xiao et al., 2023). However, the impact of aerosols on

DCSs remains poorly understood due to limited observations and complex mechanisms involved. Previous studies have found

competing effects of aerosols on clouds. On the one hand, aerosol invigoration effect shows the enhancement of convection:

increasing aerosol loading produces an increase in latent heat released by more cloud water freezing aloft due to the suppression

of collision-condensation and warm rain formation (Andreae et al., 2004; Khain et al., 2012; Koren et al., 2014). On the other



hand, the aerosol radiative effects inhibit convection, which is the aerosol that blocks sunlight from reaching the surface,

especially in a heavily polluted environment (Jiang et al., 2018). In addition, the absorptive aerosol raised by convection may

also accelerate the evaporation of cloud droplets or the sublimation of ice crystals by radiative heating (Rosenfeld et al., 2008).

This is known as aerosol semi-direct effect. And the stabilization of the temperature profile by the aerosols heating will further

inhibit convection. The effects of aerosols and meteorological factors on clouds are coupled, posing challenges to the study of

aerosol-cloud interactions. Based on cloud-resolving model, Fan et al. (2009) find that wind shear qualitatively determines

whether aerosols inhibit or enhance DCSs strength, i.e., aerosols promote convection under weak wind shear while the opposite

occurs under strong wind shear. Xiao et al. (2023) discovered that the net latent heat released by microphysical processes in

aerosol-cloud interaction can increase the vertical velocity and facilitate the development of the DCSs. Through partial

correlation analysis, Jiang et al. (2018) find that the qualitative effects of aerosols and meteorological factors on deep

convection had regional differences. That is, in some cases, aerosols and meteorology affect convection in the same direction,

but in other cases they might be opposite.

In this study, we decouple meteorology and aerosol effects by discussing the relationship between meteorological factors

and the development of DCSs under different background aerosol concentrations (Fig. 6). Sect. 2.3 provides details on the

selection and matching of aerosol data. The aerosol mass concentration is divided into polluted and clean environments using

the 30% and 70% quantile as thresholds. The impact of meteorological factors on the development of DCSs is illustrated in

Fig. S8. The width of DCSs is positively correlated with wind shear in both regions, as shown in Fig. 6a (and Fig. S8a, d).

Although the correlation between DCS width and wind shear is not significant in the TP region under low aerosol loading, this

may be due to a limited sample. Alternatively, the development of DCSs on the TP may be somewhat restricted due to a limited

amount of CCN and water vapour supply. These results indicate that wind shear qualitatively promotes the horizontal

development of DCSs. Although vertical wind shear can cause cloud particles to fall into drier air from the cloud updrafts,

leading to increased sublimation and evaporation (Khain, 2009) and reduced convection. However, based on model, Neggers

et al. (2003) found that wind shear could enhance the tilting of clouds and increase cloud fraction. The wind shear can intensify

updrafts and make cold pools colder, particularly when the shear direction reverses across the jet level (Robe and Emanuel,

2001). Fig. 6b, e (and Fig. S8b, e) shows the difference in the horizontal development of DCSs under different conditional



stability of moist convection (e.g., $\partial\theta_{es}/\partial z$). The results indicate that an increase in $\partial\theta_{es}/\partial z$ results in a wider DCS width. A larger $\partial\theta_{es}/\partial z$ implies a more stable atmosphere and a stronger inversion near the anvils of DCSs. It has been found that a

stronger inversion can reduce entrainment drying and warming, resulting in a moister atmosphere and wider DCSs (Myers and Norris, 2013). The differences in DCSs width under two aerosol loading conditions are relatively small (Fig. 6a, b, d, e), suggesting that the influence of meteorological factors on the width of DCSs possibly be dominated. In the TO, the DCSs in polluted environments are partly wider than those in clean environments. This suggests that when aerosols increase, although cloud droplets evaporation and ice crystal sublimation are accelerated due to aerosol semi-direct effects (Rosenfeld et al.,

2008), the invigoration of convection by aerosol (Andreae et al., 2004; Khain et al., 2012; Koren et al., 2014) can still lead to an increase in the width of DCSs. It is important to note that this study only analysed aerosols under the base of DCCs, whereas DCSs involve entrainment over a much larger area.

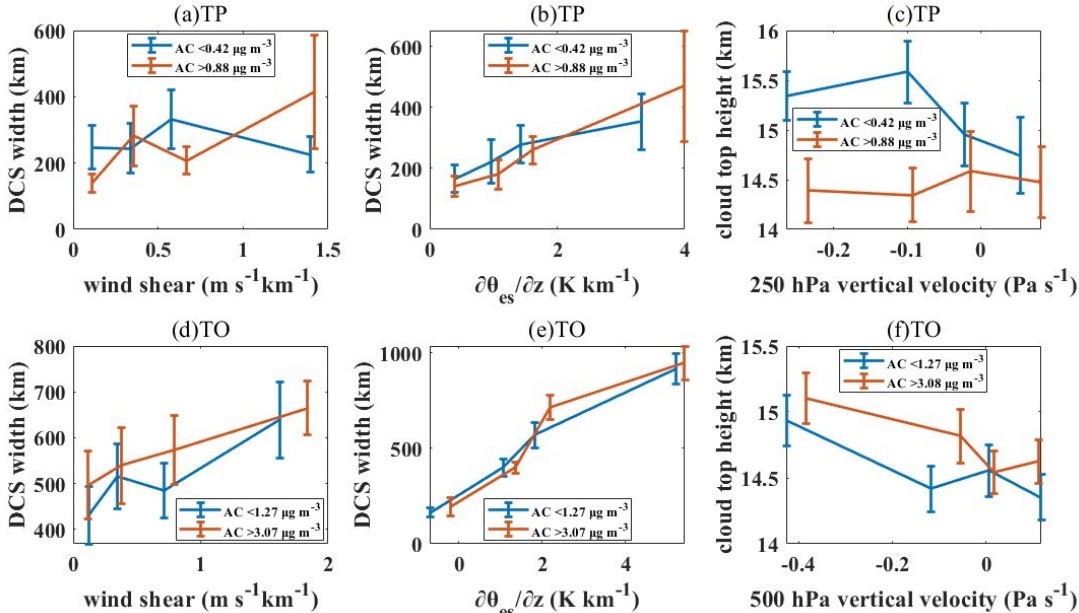

**Figure 6: Bin-averaged wind shear (m s-1km-1; a, d), the vertical gradient of the saturated equivalent potential temperature $\partial\theta_{es}/\partial z$**

**(K km-1; b, e) and vertical velocity (Pa s-1; c, f) with DCSs width (km) and cloud top height (km) from different subregions. The blue lines represent the results at weak aerosol loading (less than 0.3 quantile of the sample) and the red lines represent the results at strong aerosol loading (more than 0.7 quantile of the sample). The error bars represent the standard error of the mean (SEM=standard error / $\sqrt{n}$).**



Fig. 6 (c, f) illustrates the relationship between vertical velocity and cloud top height in different regions. The negative

value of vertical velocity indicates the updraft, while the cloud top height refers to the maximum of cloud top heights for all

profiles in each DCC. Previous studies have confirmed that upward velocity contributes to the development of deep convection

(Khain et al., 2005; Koren et al., 2010; Jiang et al., 2018). High upward motion results in efficient transport of liquid cloud

water to the level of homogeneous nucleation and longer convection lifetime (Ekman et al., 2007). Additionally, homogeneous

nucleation is more efficient with high vertical wind speed (Heymsfield et al., 2005), which is closely related to cloud dynamics,

precipitation rate and radiative properties. And as a result, more liquid water is converted to ice through homogeneous freezing,

promoting deep convection (Ekman et al., 2007). However, the cloud top height does not significantly increase with upward

velocity in the TP (Fig. S8c). This suggests that there are factors other than vertical velocity affect cloud top height of DCCs

in the TP. As shown in Fig. 6c, the correlation between vertical velocity and cloud top height is more significant under low

aerosol loading and there are distinct differences between two aerosol loading conditions. That is, the cloud top height is much

higher in clean environments in the TP. We found that the cloud base temperature of DCCs in the TP is much lower than that

in the TO (Fig. S9). In the TP, where the average altitude exceeds 3000m, the cloud base temperature of 44% of the DCSs is

below zero, despite the restriction that the cloud base of DCSs must no more than 3km from the surface in the identification.

Aerosol microphysical effects that promote the development of convection are primarily effective in warm-base clouds (>15℃)

(Rosenfeld et al., 2008; Li et al., 2011). In cold-base cloud systems, a significant amount of condensate freezes. The high

concentration of CCN impedes the autoconversion process, leading to a significant portion of cloud droplets freezing into

small ice particles without an efficient mechanism for coagulation and subsequent precipitation. These ice particles fail to

release thermal buoyancy during the processes of freezing and precipitation, requiring additional energy for ascent.

Consequently, convection is inhibited under high aerosol loading (Rosenfeld et al., 2008). While the invigoration of aerosol

microphysical effects becomes ineffective, aerosol radiative effects also inhibit the development of convection (Jiang et al.,

2018). Above discussion can interpret why the cloud top of DCSs is higher under clear air condition over the TP. In the TO,

the intensity of ascending motion showed a strong positive correlation with cloud top height (Fig. S8). Higher upward velocity

means more efficient transport of cloud water, promoting convection development (Ekman et al., 2007). Different from the





TP, the aerosol invigoration effect (Khain et al., 2005; Koren et al., 2010; Jiang et al., 2018) is effective over the TO because

the warm-base DCSs dominate in the TO (almost all of over 15 ℃), resulting in higher cloud tops under higher aerosol loading.

**3.3.2 The effect of aerosols and meteorological factors on the precipitation of DCSs**

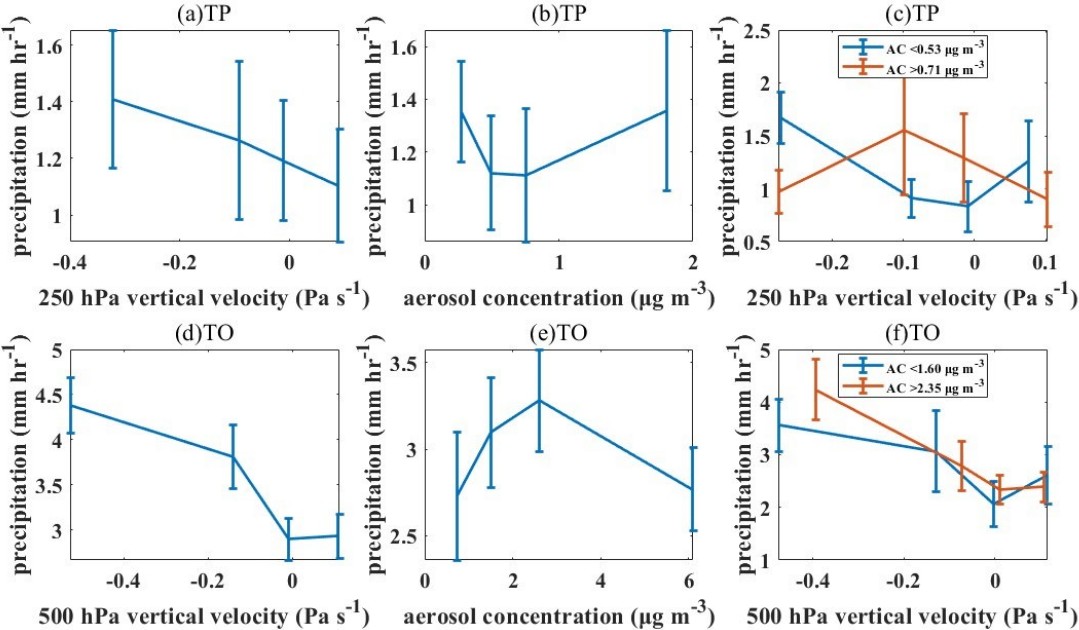

**Figure 7: Bin-averaged vertical velocity (Pa s⁻¹; a, d) and aerosol concentration (µg m⁻³; b, e) with precipitation (mm hr⁻¹) from different subregions and the relationship between vertical velocity and precipitation under different aerosol conditions (c, f). The blue lines represent the results at small aerosol loading (less than 0.3 quantile of the sample) and the red lines represent the results**
**at large aerosol loading (more than 0.7 quantile of the sample). The error bars represent the standard error of the mean (SEM=standard error / $\sqrt{n}$).**

Deep convective precipitation, as the main precipitation type in the TP (Wang et al., 2019), exerts a profound influence.

However, there has been insufficient attention paid to DCC precipitation over the TP, despite its significant effects on both the

plateau itself (Wang et al., 2019) and its downstream regions (Liu et al., 2019b). Liu et al. (2019b) found that aerosols enhance

deep convection over the TP, which then moves eastward under favourable meteorological conditions, resulting in heavy

precipitation over the Yangtze River basin and North China. In this study, we focus on the precipitation of the DCCs, which

is mainly convective precipitation (Feng et al., 2011). The relationship between precipitation and meteorological factor (i.e.,



vertical velocity) and fine aerosol concentration is explored (Fig. 7). Sect. 2.3 provides details on the selection and matching

of aerosol data. The average precipitation of different regions is presented in the last column of Table 2. The results show that,

due to the limited water vapour and convection intensity, the precipitation of the DCCs is significantly less in the TP (1.2 mm

hr$^{-1}$) than in the TO (3.5 mm hr$^{-1}$). Fig. 7 (a, d) shows the relationship between precipitation and vertical velocity for DCCs in

different regions. I-n all regions, an increase in upward motion velocity corresponds to an increase in DCCs precipitation.

Higher vertical velocity is more conducive to the transport of larger particles and/or more hydrological material, such as water

vapour, into the upper atmosphere, which in turn results in heavier precipitation (Luo et al., 2014).

The aerosol-cloud-precipitation interaction about deep convection has received much attention in recent years (Tao et al.,

2007; Li et al., 2010; Han et al., 2022). Aerosols can inhibit warm-rain processes by reducing the cloud droplet size and causing

a narrow droplet spectrum which inhibits collision and coalescence processes (Squires and Twomey, 1960; Warner and

Twomey, 1967; Warner, 1968; Rosenfeld, 1999). This leads to more and smaller cloud droplets being transported to the upper

layers to condense, resulting in more latent heat being released. This process is known as the 'aerosol invigoration effect'

(Andreae et al., 2004; Rosenfeld et al., 2008) and promotes convection and precipitation. Fig. 7 (b, e) depicts the relationship

between aerosol concentration and DCCs precipitation in different regions. Fig. 7b shows that the aerosol invigoration effect

on DCCs precipitation is not significant in the TP. One possible explanation is that the aerosols absorb solar radiation and

stabilize the daytime boundary layer, thereby inhibiting precipitation (Koren et al., 2008). Previous studies suggest that this

suppression process dominates whether the cloud is above or within the aerosol layer, even though the upper atmosphere can

be less stable due to the aerosol heating below (Koren et al., 2008). Additionally, the TP experiences a large number of cold-

base DCSs due to its high surface altitude (The cloud base temperature of DCSs in different regions is shown in Fig. S9). As

theory presented by Rosenfeld et al. (2008), clouds with cloud top temperature of less than -4 ℃ and cloud base temperature

of more than 15 ℃ are considered the most favorable conditions for the aerosol invigoration effect (Li et al., 2011). In cold-

base clouds, large amounts of condensate freeze and the presence of high concentrations of CCN can slow down

autoconversion, leading to the freezing of most cloud droplets into small ice particles propelled by the strong updraft. These

particles lack an effective mechanism to coagulate and fall as precipitation (Rosenfeld et al., 2008). And in mixed-phase/ice

clouds, the increase in soluble aerosols will lower the freezing point, inhibiting the formation of ice crystals, according to





Raoult's law (Bancroft and Davis, 1928). This theory might also explain why increased aerosols inhibit TP DCCs precipitation.

In conclusion, the precipitation of DCCs in the TP does not show a significant change with increasing aerosols probably due

to the combination of insufficient aerosol invigoration effect caused by the cold cloud base and inhibition effects from aerosols.

In the TO, there is a clear increase in DCCs precipitation with increasing aerosol concentration under clean conditions (Fig. 7e). This phenomenon can be explained by the "aerosol invigoration effect" mentioned above. However, as aerosols continue to increase, their effect on DCC precipitation shifts from invigoration to suppression. Previous studies (Dagan et al., 2015; Wang et al., 2018; Liu et al., 2019a) have also observed similar phenomena. This may be due to the fact that the aerosol

microphysical effect (or aerosol invigoration effect) is more sensitive in relative clean environments. While, under high aerosol loading conditions, the aerosol radiative effect dominates (Koren et al., 2008; Wang et al., 2018). The radiative heating of absorbing aerosols increases atmospheric stability and reduces the moisture content due to evaporation (Ackerman et al., 2000). In our results, we similarly find that the DCC under polluted conditions corresponds to a smaller upward motion velocity (Fig. S10a). The aerosol-cloud response takes on a boomerang shape when two competing effects are present (Rosenfeld et al., 2008;

Jiang et al., 2018). Nevertheless, the aerosol radiative effect over the oceans is expected to be weaker due to the ocean's high heat capacity, which prevents rapid changes in surface temperatures (Liu et al., 2019a). Meanwhile, aerosols under DCSs, which cover a large area, can only absorb a limited amount of solar radiation. This can further reduce the impact of aerosol radiative effect. Observations and cloud models over the tropical ocean demonstrate that this suppression of precipitation by aerosols under polluted environments is more likely to be due to enhanced entrainment (Liu et al., 2019a). Under the influence

of increased aerosols, cloud droplets size become smaller (Squires and Twomey, 1960; Warner and Twomey, 1967) and more susceptible to consumption by entrainment (Xue and Feingold, 2006). Furthermore, polluted conditions cause a delay in precipitation, allowing for increased evaporation and sublimation of hydrometeors in the unsaturated part of clouds, ultimately leading to a reduction in cloud water (Liu et al., 2019a). Corresponding to Liu et al., (2019a), we checked the 500 hPa relative humidity corresponding to the DCCs in the TO region (Fig. S10b) and found that it decreases as the aerosol loading increases.

This implies that the entrainment process has a greater impact under polluted conditions.

The precipitation of DCCs remains a challenging task due to the complex mechanisms involved in aerosol-cloud-precipitation interactions and variable meteorological fields. To address this issue, we attempted to differentiate between clean

and polluted environments and discuss the relationship between precipitation and meteorological factors separately. This allowed us to decouple the effects of aerosols and meteorological fields. Fig. 7 (c, f) shows the relationship between vertical

velocity and precipitation for different aerosol loadings. In Fig. 7c, the results for low aerosol loading indicate that DCCs with greater upward velocity also have more precipitation in the TP. In this case, precipitation is primarily enhanced by stronger transport of particles and hydrometeors (Luo et al., 2014). Under high aerosol loading, the varied responses of DCCs with different cloud base temperatures result in an insignificant relationship between vertical velocity and precipitation (shown by the red line in Fig. 7c). We checked the cloud base temperature of DCCs in the first bin, which are under the strongest upward

motion but do not produce much precipitation. The results show that average cloud base temperature of the DCCs in the first bin is only -1.25 ℃, which is lower than the average for the TP (0.87 ℃). This means that DCCs from the first bin may be less able to be aerosol-invigorated to produce more precipitation, and instead lead to a suppression of precipitation due to aerosol direct (Koren et al., 2008; Wang et al., 2018) and semi-direct effects (Rosenfeld et al., 2008). The results in the TO (Fig. 7f) show no significant differences between the two aerosol loading conditions, with both exhibiting a similar increase

in precipitation as the upward velocity increases. As discussed in Fig. 7e, under the influence of competing aerosol effects, increased aerosols show an invigoration followed by a shift to inhibition of precipitation. Fig. 7f demonstrates that DCCs still produce slightly more precipitation under higher aerosol loading compared to lower aerosol loading.

## 4 Conclusions and discussion

TP, as the 'Asian water tower', exerts a global-scale influence on land-ocean-atmosphere interaction and hydrological

cycle (Xu et al., 2008). Over the TP, deep convection plays a crucial role in the energy and hydrological cycle. However, previous studies on the CRE of deep convective clouds predominantly concentrate on the deep convective cores, rather than considering the combined effects of the deep convection systems, including anvils. In this study, the entire cloud cluster containing the deep convective core (DCC) and anvils is identified as the deep convection system (DCS). Using data from CloudSat and CALIPSO satellites, we explore the horizontal and vertical structure characteristics of TP DCCs/DCSs and



compare them with the results in the TO. Additionally, the radiative effects of DCCs/DCSs and the effects of meteorological

factors and aerosols on the DCCs/DCSs are discussed. The main results are as follows:

1.    Differences in terrain, water vapour, dynamical and thermal conditions between the two regions result in

varying DCS characteristics. The DCSs in the TP are less than half as wide as those over the TO, and they also exhibit a

smaller thickness.    Due to the influence of monsoon, DCSs in the TP develop most vigorously in summer, particularly

in the southeastern TP. There is no significant seasonal difference in the DCS characteristics in the TO, except for less

frequent occurrences but wider DCSs in summer.

2.    The optically thin anvils in the DCSs typically offsets some of the radiative cooling effects of the DCCs. The

DCCs in the TP are relatively thinner, resulting in weaker radiative cooling effects at the TOA compared to the TO.

Despite the thinner anvils in the TP, possibly due to more densely packed cloud tops, the radiative cooling effect of TP

anvils is stronger than compared to the TO. The average TOA CRE of TP DCSs is more negative compared to DCSs in

the TO, where the anvils are wider. The unique vertical profiles of temperature and water vapour associated with the

uplifted topography cause the DCCs/DCSs to result in more radiative LW heating at the surface and low-level atmosphere

in the TP. However, both the positive part of the LW CRH profile and the negative part of the SW CRH profile in the

low-level atmosphere are thinner, resulting in slightly smaller ATM CRE of DCCs/DCSs in the TP.

3.    Meteorological factors have significant effects on the development of DCSs and their precipitation. Both high

wind shear and $\partial\theta_{es}/\partial z$ can effectively promote the horizontal development of DCSs by enhancing cloud tilt and reducing

entrainment, respectively. High-level vertical ascent velocity is positively correlated with DCCs cloud top height in the

TO. And the cloud top height is larger under high aerosol loading due to aerosol invigoration effect. However, the

relationship between vertical velocity and cloud top height is not significant in the TP, and the cloud tops are lower under

high aerosol loading. It is because that the aerosol invigoration effect mainly occurs in warm-base convection (Rosenfeld

et al., 2008; Li et al., 2011). Therefore, the development of cold-base DCSs in the TP are inhibited under polluted

conditions, experiencing suppression of the aerosol radiative effect.

4.    Aerosols have quite different effects on precipitation in the two regions. In the TP, there is no significant

correlation between aerosol loading and precipitation because insufficient aerosol invigoration effect on cold-base DCSs.



For the warm-base DCSs in the TO, the increased aerosols have an invigoration effect on precipitation under clear conditions, but this effect shifts to suppression under polluted conditions. This is because the dominant effect transitions from aerosol invigoration to aerosol radiative effects and enhanced entrainment suppression.

        The diverse DCCs and DCSs structural features in different regions suggests the importance to explore complete cloud clusters, rather than just focusing on profiles labelled with cloud types 'deep convection' (e.g., DCC). However, this also leads

to the limitation of sample size, which prevents us from further studying the details of DCSs. To ensure statistical significance, this study does not distinguish between daytime and nighttime DCSs. This is due to the unavailability of nighttime data after 2011 caused by CloudSat battery failures. Previous studies suggest that daytime and nighttime DCSs may have different structural features or involve different processes. Rao et al. (2019) discovered distinct convective induction mechanisms that are closely linked to boundary layer wind speed during both daytime and nighttime. These mechanisms are associated with

daytime solar radiative heating and nocturnal acceleration of the low-level southwesterly wind, which is linked to the inertial oscillation. We are attempting to develop algorithms that, in combination with passive satellites, will enable the tracking of deep convective trajectories and changes in cloud properties on the plateau. This will overcome the limitation of overpass time of polar-orbiting satellites. By comprehensively monitoring the complete life cycle of DCS, we anticipate gaining enhanced insights into the underlying factors governing their behaviour and the subsequent effects downstream. This endeavour holds

the potential to furnish crucial observational evidence for refining forecasting techniques and advancing model development in the field of atmospheric science.

        In addition, in this study, we not only explore the radiative effects of the DCCs but also consider the influence of the anvils on the CRE and CRH of the DCSs. Cirrus clouds depend on deep convection, which can be directly derived from anvils or stimulated by indirect effects of deep convection (e.g. radiative cooling effects of deep convection, pileus-like updrafts, and

gravity waves which often are initiated by organized convection) (Sassen et al., 2009). In particular, we note that more than 30% of DCSs can penetrate the tropopause (shown in Table 2), which means that a large number of cirrus clouds in the stratosphere can be supplied by convection. Although our definition of DCS (The DCC cloud top must be higher than 12km, which is close to the upper troposphere) may contribute to this surprising number, it also suggests that we need to pay more attention to this phenomenon of overshooting clouds. In our previous study (Zhao et al., 2023), we conducted a quantitative

analysis of overshooting cloud cover and its diurnal variability using the cloud-aerosol transport system (CATS). Our findings

revealed that the cloud cover of overshooting clouds is higher during the night and reaches its maximum value at 16:00 UTC.

Furthermore, Zhao et al. (2024) suggested that there will be an increase in overshooting clouds in warming climates. Building

upon these findings, our future objectives involve investigatingthe general mechanisms associated with overshooting

convection, and quantifying the radiative impacts of overshooting convection from a climate-oriented perspective.

**Data availability**

2B-CLDCLASS-lidar and 2B-FLXHR-lidar data are available from CloudSat Data Processing Center:

https://www.cloudsat.cira.colostate.edu/data-products/2b-cldclass-lidar(2b-flxhr-lidar) (last access: 31 October 2022). The

IMERG datasets are available from Goddard Earth Sciences Data and Information Services Center (GES DISC):

https://doi.org/ 10.5067/GPM/IMERG/3B-HH/06 (last access: 13 December 2023). The ERA5 datasets are available from

Climate Data Store (CDS): https://doi.org/10.24381/cds.bd0915c6 (Accessed on 8 May 2023). The MERRA-2 datasets are

available from Global Modeling and Assimilation Office (GMAO): https://doi.org/10.5067/LTVB4GPCOTK2 (last access:

12 July 2023).

**Competing interests**

Some authors are members of the editorial board of journal Atmospheric Chemistry and Physics.

**Author contribution**

YZ and JL organized the paper and performed related analysis. YZ prepared the manuscript with contributions from all co-

authors. JL conceptualized the paper and revised the whole manuscript. DW, YL downloaded the data used in this study. YW

and JH provided suggestions for this study. All authors contributed to the discussion of the results and reviewed the manuscript.



## Acknowledgements

This research was jointly supported by the NSFC Major Project (42090030), the Strategic Priority Research Program of the

Chinese Academy of Sciences (XDA2006010301) and the National Science Fund for Excellent Young Scholars (42022037).

We would like to thank the CloudSat, CALIPSO, GPM, ERA5, and MERRA-2 science teams for providing excellent and

accessible data products that made this study possible.

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
