# Peer review of "Distinct structure, radiative effects and precipitation characteristics of deep convection systems in the Tibetan Plateau compared to tropical Indian Ocean"

_EGUsphere, 2024_

## Author Comment (AC1)

**Response to Reviewer #1's Comments:**

Yuxin Zhao et al. (Author)

We appreciate the constructive suggestions and insights provided by the Reviewer #1, which have helped us identify areas for improvement in our manuscript. We acknowledge that our manuscript overly elaborated on possible mechanisms in the results section, thereby detracting from the clarity of the main points and the novelty of the study. To address this, we have relocated some of the discussions of the possible mechanisms to the "Conclusion and Discussion" section. We have revised the manuscript according to the Reviewer #1's comments. Please see our point-by-point response to the comments. All revisions are shown in revised manuscript by using track changes.

**General responses:**

This is a comprehensive and robust characteristics of deep convective cloud features, radiative effects and controls over a region with particularly interesting geography. The paper is long, and it is hard to keep track of what results are novel versus recounting literature. Ideally, the paper would have been written with a clear and focused results section without referring to so much literature, and then discussed the literature in a dedicated discussion section. Broadly speaking the authors seems to confirm that the differences they see between the clouds in the two regions are consistent with understanding in the literature.

I can't say I find the paper particularly novel or easy to read. However, I can see it being a useful characteristics for other scientists to build upon. So, apart from the need to address the minor comments below, I can see value in publication.

**Response:** We greatly appreciate the reviewer's insightful comments for our work. We fully agree with the reviewer's comments. Indeed, our manuscript is very long, aiming to comprehensively analyze the structural characteristics, cloud radiative effects, precipitation, and the influences of meteorological fields and aerosols on deep convection systems across various regions. We also consider that extensive previous findings and references have potentially masked our own results in our manuscript. In

response to the reviewer's comments and suggestions, we have revised the discussion of potential mechanisms and references to literature in the "Results" section, as well as adjusted the description in the manuscript. Discussions for potential mechanisms relevant to our findings have been moved to the "Conclusion and Discussion" section. Detailed modifications can be found in the revised manuscript we have submitted.

**Minor comments**

1. Title – I haven't seen anything in the paper that makes me thing the clouds on TP are "unique", please remove the word or justify with respect to deep convective clouds in general.

   **Response:** Thank you very much for the reviewer's helpful comments. We agree that the word "unique" is too strong. So, it is changed to "distinct" in the revised manuscript. In this study, we aim to demonstrate that the different structural characteristics, cloud radiative effects, precipitation, and the impacts of meteorological fields and aerosols of deep convection systems over the TP compared with the TO. Specifically: 1) Due to the influence of terrain forcing and other factors, the DCSs over the TP are significantly thinner, but their anvils are denser (geometrically thin but with high optical depth), resulting in a stronger shortwave radiative cooling effect. 2) The dry, cold surface of the TP results in less upwelling longwave flux emitted by the surface under clear-sky condition, making the radiative heating of the DCSs more efficient at the surface. 3) Even when the cloud base of the DCS is close to the surface (distance < 3 km), the cloud base temperature over TP can still be partially below 0°C, which is rarely observed in lower-altitude regions. The colder cloud base also contributes to the unique impact of aerosols on the development and precipitation of DCS over the TP. We apologize that our initial presentation may not have effectively conveyed our viewpoint. To address this, we have added the discussion in the "Conclusion and Discussion" section. Please refer to the revised manuscript for specific modifications.

2. Title – "Tropical oceans" is much broader than the use here. At best tropical "Indian" ocean could be used.

**Response:** We appreciate the reviewer's advice. It is revised to "tropical Indian Ocean". In fact, our definition of the TO region was intended to select tropical oceanic areas adjacent to the TP while maintaining consistency with the TP region's area.

3. L19 - "competition between invigoration and radiative effects of aerosols" I think you should be specific about what radiative effect is competing with invigoration. This is a bit vague for an abstract. Perhaps "direct" radiative effect?

   **Response:** In fact, not only does the direct radiative effect (i.e., aerosols blocking incoming solar radiation at the surface) exert a suppressing influence, but the semi-direct radiative effect also does. The absorbing aerosols heat the atmosphere, enhancing its stability and consequently inhibiting convection development. Simultaneously, this process promotes the evaporation of cloud droplets, further suppressing convection (Ackerman et al., 2000). Generally, this mechanism is referred to as the aerosol semi-direct effect. To keep the abstract as clear as possible, we revise the "radiative effects" to "direct/semi-direct radiative effects".

   **See the line:** 19.

4. intro/methods – The lack of map in the main text doesn't help the reader. That's your choice, but please at least refer to figS1 early on so the reader can look at where you are studying. I also do not know the motivation behind the specific TO region you've picked, the topical ocean is much more general that box. What's special about that part?

   **Response:** Thank you very much for the reviewer's comments. To enhance the readability of the manuscript, we have moved the study area map originally presented in FigS1 to the beginning of the Methods section. Additionally, we have included a description of the study area in the Introduction section. Of course, the tropical ocean is a very large area. To make the description more precise. Following the reviewer's suggestion (Comment #2), we have revised the description of this region to the tropical Indian Ocean. We did not have any specific intention in choosing this box as the study area. We simply wanted to select a nearby ocean

region for land-sea comparison and for comparing different latitudes. We selected only the region 0°N~7°N and 68°E~93°E to maintain consistency with the TP region's area, ensuring meaningful comparisons of frequency analysis. We have added the rationale behind our partition selection in both the Introduction and Methods sections. Please refer to the revised manuscript for details.

**See the lines:** 60-61 and 99-100.

5. FigS1 – please mark on the sub-divisions of TP that you use in table 2.

   **Response:** Thank you for your suggestion. We have now marked the sub-region divisions from Table 2 in the original Fig. S1, which is now Fig. 1.

6. Section 2.2 – Have ERA5 or MERRA2 been evaluated over the tibetan plateau. This seems important. ERA5 is ultimately a model with a spatial resolution that may be affected by large gradients in terrain in the region.

   **Response:** We fully agree with the reviewer's perspective. It is necessary to assess the uncertainty of the data before using it. The high and complex topography of the Tibetan Plateau indeed induces challenges for model simulations. However, the meteorological data from ERA5 is still reliable. For example, Han et al. (2021) found in the evaluation of meteorological parameters derived from ERA5 based on radiosonde measurements on the Tibetan Plateau that ERA5 data has good reliability for atmospheric parameters in the free atmosphere. The bias and root mean square error (RMSE) for temperature are generally less than 1.2 K, and for wind speed, the bias and RMSE are generally less than 2 m/s.

   The aerosol data provided by MERRA-2 has also been evaluated over the Tibetan Plateau in the past. For example, Xu et al. (2020) found that MERRA-2 aerosol optical depth (AOD) was consistent with Aerosol Robotic Network (AERONET) and Multi-angle Imaging Spectro Radiometer (MISR) over the Tibetan Plateau. The correlation coefficients were 0.73–0.88 and 0.94, respectively. Here, AERONET provides ground-based observation data, with aerosol retrieval accuracy reaching 0.01-0.02 (Xia et al., 2004), commonly used to validate the accuracy of benchmark data from remote sensing retrievals. Of course, due to the lack of observation data over the Tibetan Plateau, reanalysis data contains a certain

degree of uncertainty Li et al. (2024). However, considering our study requires aerosol mass concentration data with complete spatial coverage and high spatial and temporal resolution, MERRA-2 is the most suitable dataset to meet this requirement.

We have added the evaluation of reanalysis data over the Tibetan Plateau in Section 2.2.

**See the line:** 157-160 and 171-173.

7. L168 – Presumably there's a third criteria that there is cloud present between the base and the top?

**Response:** We greatly appreciate the reviewer's comments. Perhaps our description was not clear enough, which caused confusion for the reviewer. However, the third criteria—cloud presence between the top and base—is not necessary. When selecting DCS samples, the cloud base height and cloud top height were obtained from the parameters "cloud base height" and "cloud top height" provided by 2B-CLDCLASS-LIDAR, which is one of cloud layer products from CloudSat and CALIPSO. The continuous vertical range of clouds is considered one layer; any interruption indicates a different layer. Therefore, the presence of clouds is guaranteed between the cloud base and cloud top, and no additional requirements are needed. Furthermore, the "bwboundaries" function was applied to verify connectivity when selecting DCS samples, ensuring that DCC also represents continuous cloud presence without interruption.

8. L179 – No "high cloud" in the image despite cloud tops over 15km. What's the definition of "high cloud"?

**Response:** The "high cloud" refers to cirrus, cirrocumulus and cirrostratus. The classification is from 2B-CLDCLASS-LIDAR datasets, based on cloud height and phase, maximum effective radar reflectivity factor, and temperature, as well as the presence of precipitation reaching the surface. The detailed cloud features for "high cloud" are as follows: (1) cloud base height more than 7 km. (2) no rain. (3) horizontal dimension is 1 to $10^3$ km. (4) vertical dimension is moderate. (5) liquid

water path (LWP) = 0.

"high cloud (cirrus, cirrocumulus and cirrostratus)" is added in the revised manuscript.

**See the line:** 109.

9. Eq1 – Another way to look at this is the abs(dV/dz). I think that's a bit more intuitive to understand. It does mean that you can identify high shear as a result of strong low level winds, and weak upper level winds. Does this occur in your data? Is it you intension to include such conditions? Do you think strong low winds with weak upper winds is likely to have the same the effect as weak low winds and strong upper winds?

**Response:** In our manuscript, the wind shear does not differentiate between the relative magnitudes of the upper and lower wind speeds. Instead, it is represented directly by the absolute value of the wind speed difference between the two layers. If there is a wind speed difference between the two layers of the atmosphere, it indicates relative motion between them, which is conducive to horizontal cloud development. This method of representing wind shear using the absolute value of the wind speed difference is commonly used in previous studies (e.g., Sherwood and Wahrlich, 1999; Naud et al., 2008). To address the reviewer's concerns, we removed the step of taking the absolute value and simply calculated dV/dz. The results are shown in Fig. R1.

The results show that the upper-layer wind is stronger than the lower-layer wind in most samples, although there are cases where the lower-layer wind is stronger (see the text annotation in Figure R1). Examining the relationship between wind shear and DCS width under different conditions, we find that larger wind shear promotes the horizontal development of DCS, regardless of whether the upper-layer wind or lower-layer wind is stronger. In the TP region, the results for cases with stronger lower-layer wind are not significant, likely due to the small sample size, whereas in the TO region with more samples, the relationship is more significant. Based on the above, we believe it is unnecessary to distinguish between the relative magnitudes of the upper and lower winds in the calculation of wind shear. To

ensure an adequate sample size, we use abs(dV/dz) to represent wind shear. In reference to the reviewer's comments, we have revised the wind shear expression to abs(dV/dz). And we added the description of this method in the revised manuscript.

**See the lines:** 212-213 and 216-217.

[Figure]

**Figure R1: Bin-averaged wind shear (dV/dz; m s$^{-1}$km$^{-1}$) with DCSs width (km) from different subregions. The error bars represent the standard error of the mean (SEM=standard error /$\sqrt{n}$).**

10. L195 – please can you describe the theoretical basis for using the gradient in theta_es to study the impact of conditional instability. I would have expected you would relate the environmental temperature to theta_es to look at stability. And for conditional stability you would need to consider the dry adiabat too (see AMS glossary on "conditional stability"). I did look at Li et al (2018) but I found no information to justify the approach.

**Response:** Thank you very much for the reviewer's comments. The theta_es is the "saturated equivalent potential temperature" or "pseudo-equivalent potential temperature". The theta_es is the potential temperature that takes into account the water vapor mixing ratio. For saturated air layer, the following formula can be derived:

$$\frac{\partial \theta_{se}}{\partial z} \approx \frac{\theta_{se}}{T}(\gamma_s - \Gamma)$$

where $\gamma_s$ is the moist adiabatic lapse rate, and $\Gamma$ is environmental lapse rate. When $\frac{\partial \theta_{se}}{\partial z}$ is positive, $\gamma_s$ is larger than $\Gamma$, which means the atmosphere is stable. And the larger the $\frac{\partial \theta_{se}}{\partial z}$, the more stable the atmosphere. We take $\frac{\partial \theta_{se}}{\partial z}$ as diagnostic of the degree of atmospheric instability.

In response to the reviewer's suggestion, we have rechecked the glossary of conditional stability. Conditional instability refers to the environmental lapse rate lying between the dry and moist adiabatic lapse rate. It means that the air layer is stable for unsaturated (clear) air parcels and unstable for saturated (cloudy) air parcels. In fact, when $\frac{\partial \theta_{se}}{\partial z} > 0$, the air layer is absolutely stable. This description "conditional instability" is inappropriate here. It is revised to "atmospheric stability".

Regarding the reviewer's mention of "relate the environmental temperature to theta_es to look at stability", we wonder if it meant that the reviewer suggest that we calculate the environmental lapse rate. In fact, the calculation of theta_es requires the environmental temperature, meaning theta_es already include information about the environmental temperature. Under moist adiabat, the theta_es is constant at different altitudes. Therefore, its gradient can describe the relative magnitude of the environmental lapse rate compared to the moist adiabatic lapse rate.

Consequently, the vertical gradient of theta_es can characterize the stability of atmosphere. The gradient of theta_es has been widely applied as a criterion for atmospheric instability. For instance, it has been shown to have a significant impact

on cloud overlap (Naud et al., 2008) and serves as a predictor for tropical cyclone occurrence (McDonnel and Holbrook, 2004). McDonnel and Holbrook (2004) indicated that the gradient of theta_es is a measure of the potential for cumulonimbus convection from a lapse-rate stability viewpoint.

In summary, the gradient of theta_es is a typical proxy for atmospheric stability, widely used for convective condition analysis. It can be employed in our study to describe the thermodynamic state influencing the development of deep convective clouds. For clearer expression, we have revised the description of theta_es in the revised manuscript.

**See the lines:** 218-222.

11. Results section – this includes a lot of references and discussion for results. I suggest just not labelling those sections as "results" or calling them "results and discussion".

    **Response:** Thank you for the reviewer's suggestion. Extensive discussion and citations might have interfered with our presentation of the results. We do not change the title of the results section, but we remove some of the references and move some discussions to the discussion section.

12. Table3 – I'm surprised by how negative these CRE's are, I think this is because they're daytime-only? I think it would be worth labelling them as such in the caption.

    **Response:** Yes, the data for the shortwave cloud radiative effect only comes from daytime observations, around 13:30 local time. The shortwave cloud radiative effect observed at this time, when solar radiation is nearly at its peak, is very strong. In our study, the calculation of CRE is based on profiles where DCS/DCC are present, rather than being weighted by cloud cover over an area (or grid point), which represents the radiative effect of clouds over a large region (e.g., L'Ecuyer et al., 2019). Therefore, our CRE results appear predominantly negative. Different methods and results reflect varying perspectives in consideration.

13. L358 – fig S5 i and j are BOA DCS not ATM DCC.

    **Response:** It is corrected to "Fig. S5m" in the revised manuscript.

14. Fig6 – aerosol quantiles. I suggest you refer to 30th and 70th percentiles, opposed to the numbers you put into a code function. How do the actual values of these percentiles relate to the low and high aerosol environments discussed by Fan (are you actually spanning the range of aerosol levels they did?)

    **Response:** Thank you very much. We have revised the captions of Fig. 6 and Fig. 7 according to the reviewer's suggestions.

    The response to the question is as follows: Our study uses a different method for classifying clean and polluted environments compared to Fan et al., (2009). We match the aerosol mass concentrations provided by MERRA-2 with the observed DCS samples, and then classify the data into relatively clean and relatively polluted conditions based on the 30th and 70th percentiles. In contrast, Fan's research classifies clean and polluted conditions in numerical experiments using cloud droplet number concentrations (or cloud condensation nucleus concentration) ranging from 110 to 1100 cm$^{-3}$. Clean and polluted conditions are relative to different regions. For example, in the TP region, an aerosol mass concentration of 0.88 is considered relatively polluted, whereas this value would still be considered clean in the TO region.

15. L590 – do you mean previous studies over the TP? If so, be specific. Plenty of studies over other areas have studied the anvil CRE.

    **Response:** Yes, we mean that a lot of studies of deep convective clouds over the TP focus on deep convective cores. "previous studies on the CRE of deep convective clouds over the TP" is added in the revised manuscript.

**Technical comments**

16. L14 – "notable" to "notably"

    **Response:** It is corrected in the revised manuscript.

17. abstract – generally the abstract will need a grammar check by the journal. Issues are very minor.

**Response:** We sincerely appreciate the reviewer's valuable comment. Upon careful consideration, we have thoroughly reviewed the grammar of the abstract and addressed the issues accordingly.

18. L26 – "convections" is not grammatically correct. Change to something like "convective storms"?

**Response:** It is corrected in the revised manuscript.

**See the line:** 26-27.

19. L369 – Can you spell out in your methods the equations for each of the atmos/BOA/TOA CREs. You're saying "difference between all-sky and clear-sky" but it's ambiguous which way you have done the subtraction. Please spell it out so it is clear. I assume clear minus all-sky, but the phrasing suggests the opposite to me.

**Response:** For atmos/BOA/TOA CREs, the CREs are defined as:

$$\text{CRE}_{net} = \left(F_{net}^{down} - F_{net}^{up}\right)^{all-sky} - \left(F_{net}^{down} - F_{net}^{up}\right)^{clear}, \tag{1}$$

where $F_{net}^{down}$ and $F_{net}^{up}$ are the downward and upward net fluxes, respectively.

$$F_{net} = F_{SW} + F_{LW}, \tag{2}$$

where $F_{SW}$ and $F_{LW}$ are the shortwave and longwave fluxes.

At the TOA,

$$\left(F_{net}^{down}\right)^{all-sky} = \left(F_{net}^{down}\right)^{clear}, \tag{3}$$

Thus, Eq. (1) changes to:

$$\text{CRE}_{net} = \left(F_{net}^{up}\right)^{clear} - \left(F_{net}^{up}\right)^{all-sky}, \tag{4}$$

At the BOA and ATM, the CRE is calculated with Eq. (1).

The TOA CRE and BOA CRE are directly provided by 2B-FLXHR-lidar, and the calculation of ATM CRE is TOA CRE minus BOA CRE (Lv et al., 2015).

Cloud radiative heating rate (CRH) is equivalent to the ATM CRE per unit mass. Similarly, CRH is defined as the differently between the all-sky radiative heating rates and the clear-sky radiative heating rates (Haynes et al., 2013):

$$\text{CRH} = \text{heating rate}^{all-sky} - \text{heating rate}^{clear}, \tag{5}$$

This Eq. (5) is added in the revised manuscript.

20. L648 – "investigatingthe" needs a space

   **Response:** It is corrected in the revised manuscript.

**Reference**

Ackerman, A. S., Toon, O. B., Stevens, D. E., Heymsfield, A. J., Ramanathan, V., and Welton, E. J.: Reduction of Tropical Cloudiness by Soot, Science, 288, 1042-1047, https://doi.org/10.1126/science.288.5468.1042, 2000.

Han, Y., Yang, Q., Liu, N., Zhang, K., Qing, C., Li, X., Wu, X., and Luo, T.: Analysis of wind-speed profiles and optical turbulence above Gaomeigu and the Tibetan Plateau using ERA5 data, Monthly Notices of the Royal Astronomical Society, 501, 4692-4702, https://doi.org/10.1093/mnras/staa2960, 2021.

Haynes, J. M., T. H. Vonder Haar, T. L'Ecuyer, and D. Henderson: Radiative heating characteristics of Earth's cloudy atmosphere from vertically resolved active sensors, Geophys. Res. Lett., 40, 624–630, https://doi.org/10.1002/grl.50145, 2013.

L'Ecuyer, T. S., Hang, Y., Matus, A. V., and Wang, Z.: Reassessing the Effect of Cloud Type on Earth's Energy Balance in the Age of Active Spaceborne Observations. Part I: Top of Atmosphere and Surface, J. Clim., 32, 6197-6217, /https://doi.org/10.1175/JCLI-D-18-0753.1, 2019.

Li, W., Wang, Y., Yi, Z., Guo, B., Chen, W., Che, H., and Zhang, X.: Evaluation of MERRA-2 and CAMS reanalysis for black carbon aerosol in China, Environmental Pollution, 343, 123182, https://doi.org/10.1016/j.envpol.2023.123182, 2024.

Lv, Q., Li, J., Wang, T., and Huang, J.: Cloud radiative forcing induced by layered clouds and associated impact on the atmospheric heating rate, J. Meteorol. Res., 29, 779-792, https://doi.org/10.1007/s13351-015-5078-7, 2015.

McDonnell, K. A. and Holbrook, N. J.: A Poisson regression model approach to predicting tropical cyclogenesis in the Australian/southwest Pacific Ocean region using the SOI and saturated equivalent potential temperature gradient as predictors, Geophysical Research Letters, 31, https://doi.org/10.1029/2004GL020843, 2004.

Naud, C. M., Del Genio, A., Mace, G. G., Benson, S., Clothiaux, E. E., and Kollias, P.: Impact of Dynamics and Atmospheric State on Cloud Vertical Overlap, Journal of Climate, 21, 1758-1770, https://doi.org/10.1175/2007JCLI1828.1, 2008.

Orsolini, Y., Wegmann, M., Dutra, E., Liu, B., Balsamo, G., Yang, K., de Rosnay, P., Zhu, C., Wang, W., Senan, R., and Arduini, G.: Evaluation of snow depth and snow cover over the Tibetan Plateau in global reanalyses using in situ and satellite remote sensing observations, The Cryosphere, 13, 2221-2239, https://doi.org/10.5194/tc-13-2221-2019, 2019.

Sherwood, S. C. and Wahrlich, R.: Observed Evolution of Tropical Deep Convective Events and Their Environment, Monthly Weather Review, 127, 1777-1795, https://doi.org/https://doi.org/10.1175/1520-0493(1999)127<1777:OEOTDC>2.0.CO;2, 1999.

Xia, X. A., Chen, H. B., Wang P. C.: Validation of MODIS aerosol retrievals and evaluation of potential

cloud contamination in East Asia, J. Environ. Sci. (China), 16, 832-837, https://doi.org/10.3321/j.issn:1001-0742.2004.05.028, 2004.

---

## Author Comment (AC2)

**Response to Reviewer #2's Comments:**

Yuxin Zhao et al. (Author)

We are deeply grateful to Reviewer #2 for the suggested revisions to our manuscript. The issues raised highlighted some shortcomings in our manuscript, which we have subsequently addressed. Our conclusions regarding the different impacts of aerosols on cold and warm cloud bases was formed after analyzing observational data and considering mechanisms proposed in previous studies. Following the reviewer's advice, we revised the description in "Conclusions and Discussions" to clarify the potential mechanisms and acknowledge the uncertainty in the causal relationship between results and mechanisms. Additionally, in response to the reviewer's suggestion to average the instantaneous observations of cloud radiative effect throughout the daytime, we processed the data using the recommended method and included the results in our response (the response to Comment #15). In our understanding, this method is reliable to analyze daily mean cloud radiative effects from a climatological perspective. The method is based on the premise that clouds remain unchanged throughout the daytime, with only solar radiation exhibiting periodic variations. However, although our study is based on over a decade of satellite observations, the total sample size is only a few hundred. Hence, our results must be considered cautiously in the sense of climatology. Consequently, we revised the vague expression "daytime" and "nighttime" to "~1:30 p.m." and "~1:30 a.m." separately, emphasizing the "instantaneity". And the discussion of the limitations of daily-twice measurements is added. Please see our point-to-point reply to comments. All revisions were shown in revised manuscript by using track changes.

**General Comments**

The focus of this study was to capture deep convective system properties over the Tibetan Plateau using satellite remote sensing observations. This research is deemed novel as not much is understood of the cloud structure and radiative effects of full deep

convective systems over this region. Furthermore, it is challenging to decouple the relative influences of meteorological conditions and aerosol concentrations on the deep convective cloud structure and precipitation. Therefore, the analysis also investigated how dynamical properties such as vertical wind shear, convective instability, and vertical velocity influence cloud and precipitation development under differing aerosol loading environments. The results were supported by explanations of potential mechanisms that were theorized in previous work. However, the exact mechanisms, such as how the aerosol invigoration effect differs between warm-base and cold-base clouds, or how aerosol invigoration and aerosol radiative effects individually influence entrainment suppression, were not themselves tested or observed in this study. While it seems appropriate to speculate the potential mechanisms, the conclusions heavily relied on such mechanisms to explain the results. Therefore, I suggest that it would serve the paper better if the concluding remarks pointed to such mechanisms without stating that these mechanisms are the reason for the results.

**Response:** We appreciate the constructive suggestions provided. Indeed, our conclusions remain speculative, drawing on mechanisms proposed in previous studies. We add a description of these limitations in the "conclusions and discussion" Section. According to the reviewer's comments, we revise the descriptions about the possible mechanisms of the different effects of aerosols on warm-base and cold-base clouds in the conclusion. We are grateful for the reviewer's suggestion, which are very helpful in improving the rigor and readability of this paper.

**Specific Comments**

1. *L41*: What did Luo et al. (2011) find?

   **Response:** Luo et al. (2011) found that deep convection core over the TP is shallower, less frequent, and embedded in smaller-size convection systems. And the cloud tops of deep convection cores are more densely packed. We have

incorporated the more detailed conclusions mentioned above into the citation of Luo et al. (2011) in the revised manuscript.

**See the line:** 42-43.

2. *L70-76*: These sentences are the start of a different point, so I think they would best be served in a different paragraph that discusses how spaceborne measurements-- and particularly the ones that you are using--have been used for aerosol-cloud-precipitation measurements.

**Response:** Thank you for your suggestion. We have moved this part to a new paragraph following this section to describe the application of satellite measurements in cloud, aerosol and precipitation.

**See the lines:** 82-90.

3. *L106-107*: Measurements are two times a day at 1:30 am/pm LST, not for the full day, between 2006-2011.

**Response:** We appreciate the reviewer pointing out the imprecise description in our manuscript. In the revised manuscript, it has been corrected to: "the 2B-CLDCLASS-LIDAR data is only available at ~1:30 a.m./p.m. local time from 2006 to 2011 and solely ~1:30 p.m. from 2012 to 2019."

**See the lines:** 114-115.

4. *L111-112*: "It is important to note that CloudSat will no longer operate during nighttime due to the battery anomaly (Witkowski et al., 2018)" seems out of place. Is this in reference to the DO-Op switch made in 2012?

**Response:** Yes, we are indeed referring to the switch to DO-Op. We have revised the inappropriate description pointed out by the reviewer to the following sentence:

"It is important to note that all CloudSat data since the switch to DO-Op in late 2011 have been daytime-only (at ~1:30 p.m.)."

**See the lines:** 113 and 119-120.

5. *L115*: Is daytime SW CRE normalized to account for variability in solar insolation throughout the day? If not, that could drastically impact the results — CRE would be much more enhanced at 1:30 pm compared to the rest of the day.

**Response:** In this study, the SW CRE has not been normalized and only represents the instantaneous radiative effects observed at 1:30 p.m. As described at the beginning of our response, we think that the normalization method, which assumes clouds remain unchanged during daytime, is more applicable to climatological studies with large sample sizes. However, this method seems unsuitable for our study, which involves only a few hundred cloud clusters. In the revised manuscript, we have added the description of the limitations of instantaneous radiative effect observations. Additionally, we also attempted to normalize the SW CRE following the reviewer's suggestion and presented the results in our response to comment #15, though we did not include it in the main text.

**See the line:** 123-126.

6. *L177*: So anvil can be precipitating or non-precipitating?

**Response:** Generally, the thicker parts of the anvil near the core can be precipitating, while the thinner cirrus parts at the edges are non-precipitating. For example, as shown in Figure 1, several cumulus profiles near DCC are observed. These profiles may exist drizzle or snow. However, most of the anvil profiles are non-precipitating. As results, the precipitation from anvils are not included in the results in our study. The precipitation in our study is the precipitation from DCC.

7. *Figure 1*: Just to clarify, is this example not included in your analysis of CRE since you are ignoring systems with multiple cloud-layer profiles (i.e., systems with underlying low-level cloud)?

   **Response:** The example shown in Figure 1 is included in the CRE analysis. Although we excluded multilayer cloud profiles in DCS to eliminate their influence on the DCS radiative effect, we did not exclude the entire DCS. In other words, the CRE of the DCS shown in Figure 1 is calculated as the average CRE of its single-layer cloud profiles. We represent the average cloud radiative forcing of the entire DCS using the average results of these single-layer cloud profiles. This approach is because DCS, as a cloud system with a large horizontal extent, rarely consists entirely of single-layer clouds. Therefore, in calculating the CRE of DCS that includes multilayer cloud profiles, we exclude only the multilayer cloud profiles and use the results of the single-layer cloud profiles from that DCS. In the revised manuscript, we have improved the description of the method used for excluding multilayer cloud profiles: "In other words, in calculating the CRE of DCS that includes multi-layer cloud profiles, we average the CRE of the single-layer cloud profiles in this DCS and ignore the multi-layer cloud profiles (the results of this DCS are not excluded)."

   **See the lines:** 128-131.

8. *L208-209*: What is your motivation for selecting meteorological factors the hour before the DCS was detected by CloudSat/CALIPSO? Do you want to select them before convection is initiated, or before the DCS advects into that region? Or to match up with the aerosol information?

   **Response:** The influence of meteorological factors on the development of deep convective clouds (DCS) exhibits a lag. In other words, the state of the DCS at the time of observation is more closely related to the meteorological factors from before that moment. Therefore, we analyzed the meteorological field from one hour

prior to the DCS observation. And the one-hour movement distance of DCS relative to the ERA5 grid size does not have a significant impact (details shown in the response to Comment #9). This matching method is commonly used in studies analyzing the impact of thermal and dynamic factors on clouds and precipitation (e.g., Sun et al., 2023). The aerosol information is matched to the DCS in time to account for the effects of wet scavenging, hence the choice of aerosol information from before the precipitation event. Consequently, the selected aerosol and meteorological factor data may not be from the same time. However, when precipitation does not occur, aerosol changes are relatively minor. Consequently, the impact of this temporal matching on the results is smaller compared to wet scavenging.

9. *L209-210*: When you say that the convective system movement under advection is ignored, what are you suggesting? Are environments fairly homogeneous such that you do not need to consider the meteorological conditions in the region that convection moves into?

**Response:** I'm sorry for any confusion our expression may have caused the reader. "the convective system movement under advection is ignored" means that the one-hour movement distance of DCS relative to the ERA5 grid size does not have a significant impact. Due to the lag effect of meteorological factors on clouds, we used meteorological data from the hour before the DCS was detected by CloudSat/CALIPSO. For spatial matching, we selected meteorological data spatially proximate to the DCS/DCC. Considering the grid size of the meteorological data is 0.25°*0.25° (longitude*latitude), the DCS is unlikely to move out of its located grid during this hour, or it may move to an adjacent grid, but not far enough to experience significant differences in meteorological conditions. Referencing the study by Sherwood and Wahrlich (1999) on tropical ocean convective clouds, they found that the movement speed of convective cloud systems correlates highest with 700 hPa wind speed. We also examined the 700

hPa wind speeds corresponding to our TO region samples. Only 11.7% of the samples had 700 hPa horizontal wind speeds exceeding 10 m/s (samples that moved more than the diagonal length of a 0.25° grid in one hour). Previous studies analyzing the influence of meteorological fields on clouds or precipitation have also commonly used the method of spatial proximity with a temporal lead for matching. For example, in Sun et al. (2023), the CAPE and wind shear used are before precipitation start time in the analysis of the impact of meteorological factors on precipitation. In the revised manuscript, we have modified the expression.

**See the lines:** 235-239.

10. *Section 3.1*: Is this section considering both daytime and nighttime DCSs?

**Response:** Yes, in the statistics presented in Section 3.1, we have included both daytime and nighttime DCS without making any distinction between them. In the revised manuscript, the differences in daytime and nighttime DCSs are discussed in reference to the Comment #12.

11. *L230*: Can one cloud cluster or DCS contain multiple DCCs in your analysis?

**Response:** Yes, in our study, there are samples where DCC profiles within a DCS are discontinuous, or in other words, multiple DCCs exist. Additionally, we examined some samples where the DCC profiles were not adjacent. We found that in some cases, there were only a few interrupted profiles (the cloud type of these profiles are not deep convection) between DCCs, and these interrupted profiles exhibited fairly thick clouds. This could be due to precipitation events or other factors preventing these profiles from meeting the thresholds defined for deep convection in 2B-CLDCLASS-LIDAR. We do not count the DCCs within the same DCS separately but consider them together. For example, if a DCS contains 50 DCC profiles, even if they are not contiguous, such as 20 DCC profiles followed by a gap and then another 30 DCC profiles, the width of the DCC in this DCS is

considered as 50*1.1 km (the horizontal spacing of the profiles). When matching meteorological factors at the locations of DCCs in Section 3.3, the average of the meteorological factor data at the locations of all DCC profiles within a DCS will be matched to that DCS, even if the DCC profiles are interrupted.

12. *L232*: It should also be discussed what the differences in convection between land and ocean are, particularly with respect to the diurnal cycle. Since land and ocean have different diurnal cycles of convection and precipitation, and that CloudSat-CALIPSO are confined to only twice-daily measurements that do not capture the full diurnal cycle, how might this influence the results that you are seeing? For example, how might the diurnal cycle be influencing the differences in the frequency and structure of convection between TO and TP?

**Response:** Thank you very much for the reviewer's helpful comments. The different diurnal cycles of convection between land and ocean is indeed a noteworthy topic. Due to the CloudSat switch to DO-Op in the late 2011 and only daytime (~1:30 p.m.) data is available after 2011. We recalculate the spatial statistics of daytime (~1:30 p.m.) and nighttime (~1:30 a.m.) DCSs based on 2B-CLDCLASS-LIDAR from 2006 to 2011. The results are as follows:

| Region | Sample number | Width of DCSs (km) / SD | Width of DCCs (km) / SD | Width of anvil (km) / SD |
|---|---|---|---|---|
| TO | 285 | 612.4/564.6 | 54.0/62.6 | 558.4/542.3 |
| TP (total) | 111 | 201.4/192.4 | 21.3/15.5 | 180.1/186.3 |
| TP (NW) | 10 | 198.2/133.2 | 14.6/5.8 | 183.6/132.1 |
| TP (NE) | 18 | 136.2/148.0 | 21.6/18.0 | 114.6/134.3 |
| TP (SW) | 35 | 225.3/220.4 | 22.2/15.4 | 203.1/215.7 |
| TP (SE) | 48 | 209.1/195.2 | 22.0/16.0 | 187.1/188.7 |

| Region | DCCs /DCSs[a] (%) | Thickness of DCCs (km) / SD | DCCs/DCSs penetrating tropopause (%) | Mean precipitation of DCCs (mm hr$^{-1}$) |
|---|---|---|---|---|
| TO | 14.5 | 14.4/1.3 | 31.6/45.3 | 3.4 |
| TP (total) | 18.3 | 9.7/1.3 | 14.4/26.1 | 0.9 |
| TP (NW) | 14.0 | 9.1/1.3 | 20.0/20.0 | 0.2 |
| TP (NE) | 22.5 | 9.9/1.3 | 22.2/27.8 | 1.1 |

| Region | DCCs /DCSs[a] (%) | Thickness of DCCs (km) / SD | DCCs/DCSs penetrating tropopause (%) | Mean precipitation of DCCs (mm hr⁻¹) |
|---|---|---|---|---|
| TP (SW) | 18.6 | 9.7/1.4 | 14.3/25.7 | 0.6 |
| TP (SE) | 17.6 | 9.6/1.3 | 10.4/27.1 | 1.0 |

**Table R1: The spatial statistics of daytime (~1:30 p.m.) DCSs during 2006-2011 in different subregions. The definition of different parts of TP are as follows: TP(NW) (33.5°N–37°N, 78°E–90.5°E); TP(NE) (33.5°N–37°N, 90.5°E–103°E); TP(SW) (30°N–33.5°N, 78°E–90.5°E); TP(SE) (30°N–33.5°N, 90.5°E–103°E). SD is an abbreviation for standard deviation.**

| Region | Sample number | Width of DCSs (km) / SD | Width of DCCs (km) / SD | Width of anvil (km) / SD |
|---|---|---|---|---|
| TO | 357 | 725.4/592.6 | 72.1/73.8 | 653.3/574.5 |
| TP (total) | 10 | 974.3/856.0 | 37.0/30.4 | 937.3/865.8 |
| TP (NW) | 0 | | | |
| TP (NE) | 0 | | | |
| TP (SW) | 3 | 775.1/719.8 | 45.5/8.3 | 729.7/723.3 |
| TP (SE) | 7 | 1059.6/947.7 | 33.3/36.3 | 1056.3/958.8 |

| Region | DCCs /DCSs[a] (%) | Thickness of DCCs (km) / SD | DCCs/DCSs penetrating tropopause (%) | Mean precipitation of DCCs (mm hr⁻¹) |
|---|---|---|---|---|
| TO | 16.5 | 14.7/1.5 | 40.6/54.9 | 4.0 |
| TP (total) | 9.2 | 11.4/1.9 | 20.0/30.0 | 2.0 |
| TP (NW) | | | | |
| TP (NE) | | | | |
| TP (SW) | 19.0 | 13.4/1.1 | 33.3/66.7 | 3.3 |
| TP (SE) | 5.0 | 10.5/1.5 | 14.3/14.3 | 1.4 |

**Table R2: The spatial statistics of nighttime (~1:30 a.m.) DCSs during 2006-2011 in different subregions. The definition of different parts of TP are as follows: TP(NW) (33.5°N–37°N, 78°E–90.5°E); TP(NE) (33.5°N–37°N, 90.5°E–103°E); TP(SW) (30°N–33.5°N, 78°E–90.5°E); TP(SE) (30°N–33.5°N, 90.5°E–103°E). SD is an abbreviation for standard deviation.**

In terms of frequency, there are more DCSs occur at ~1:30 a.m. over the TO, whereas DCSs over the TP primarily occurs during daytime (~1:30 p.m.). The difference in DCS daytime and nighttime frequency is quite pronounced between the TP and the TO. However, due to the limited number of nighttime DCSs over the TP (only 10), the representativeness of their width, thickness, and precipitation statistics is very limited. The results show that, compared to daytime, nighttime DCSs on the TP are wider, thicker, and have more precipitation. On the contrary,

these parameters are greater during nighttime than during daytime in the TO. As an aside, the total sample size of the results in Table R1 and R2 is slightly larger than the sample size of results in Section 3.2 (TP-116; TO-623, shown in L115 in the manuscript). For several DCS samples, the corresponding 2B-FLXHR-lidar data files could not be available.)

Since the only twice-daily measurements CloudSat and CALIPSO cannot capture the full diurnal cycle, the different diurnal cycles of DCS in the TP and TO regions are bound to affect the results to some extent. Previous studies found that the occurrence frequency of deep convection reach the daily maximum at around 16:00-18:00 LT (Devasthale and Fueglistaler, 2010; Xu and Zipser, 2011). Shown in the results based on ISCCP brightness temperature in Kottayil et al. (2021) (Fig. R1), the diurnal peak time of deep convection occurrence frequency over Indian ocean occurs between ~5:00 and ~12:00 LT. And they found that the diurnal amplitude is smaller at sea than over land. Although the subjects of the referenced studies do not exactly match the DCS in our study, the diurnal cycle results of deep convection can provide some degree of reference. Overall, the diurnal cycle of convection has a greater impact on the twice-daily measurements CloudSat and CALIPSO results in the TP region due to the larger diurnal amplitude over land. The analysis of difference in DCS daytime and nighttime frequency is added in "Results (Section 3.1)" and the limitations of twice-daily measurements are added in "Conclusions and discussion".

**See the lines:** 266-277 and 659-666.

[Figure]

**Figure R1: The Fig. 3 in Kottavil et al. (2021). Diurnal peak time for deep convection in local time. The unit is in hours.**

13. *L248-250*: Why does deep convection over the TP contribute so much stratospheric pollutants?

**Response:** The tropics are far from the main sources of biomass and biofuel burning pollutants. And the pollutants have a strong sink from contact with the ocean. The results based on satellites observations show that much of the air in the tropical upper troposphere is relatively depleted in hydrogen cyanide. For these reasons, although the transport of air from the troposphere to the stratosphere occurs primarily in the tropics, associated with the ascending branch of the Brewer-Dobson circulation, the tropical upwelling cannot be the main source. The Tibetan Plateau is the heat source of Asian summer monsoon, and the strong convergence near the surface in summer is conducive to the transport of pollutants. And the Asian summer monsoon circulation contains a strong anticyclonic vortex in the upper troposphere and lower stratosphere. As evidenced by satellite observations, a mean upward circulation on the eastern side of the anticyclone extends the transport into the lower stratosphere. The cross-tropopause high mixing ratio of hydrogen cyanide near $30^{\circ}$ N also show the strong transport of pollution over the

Tibetan Plateau. The above evidence and analysis are from Randel et al. (2010). To clarify, we revised the expression.

**See the lines:** 290-293.

14. *Table 2*: are these exclusively single-cell DCSs? Also, I would suggest switching the order of the TP (total) and TO so that the TP regions are consecutive.

**Response:** In our study, we did not distinguish between single-cell and multi-cell DCSs. The DCSs we used include both cases. If discontinuous DCC profiles appear within the same DCS, they will be integrated for analysis together. For the calculation method of DCC statistics in multi-cell DCSs, please refer to our response to Comment #11. Currently, our differentiation of components (DCC and anvils) within DCS is based on whether they are identified as deep convection in 2B-CLDCLASS-LIDAR, which is a relatively simple criterion. We have not conducted detailed research into whether DCS are single-cell. This may require future study to utilize additional auxiliary data and conduct more in-depth investigations. We appreciate the reviewer's suggestion, and we have swapped the order of TP (total) and TO in Table 2 in the revised manuscript.

15. *L302*: How are you getting these LW and SW CRE values? The SW CRE is from daytime-only at the 1:30 pm overpass, correct? And are you averaging over the daytime and nighttime overpasses for the LW CRE? Since the SW CRE is calculated from radiative fluxes at 1:30 pm local solar time, these measurements are more enhanced than at other times of day due to the near peak in solar insolation at this time. To capture cloud radiative effects that are more representative of what they would be throughout the day, you would need to multiply the SW radiative fluxes by the diurnally averaged insolation for that day and location (L'Ecuyer et al., 2019) and then recalculate the net CRE.

**Response:** The LW CRE value is from averaging both 1:30 p.m. and 1:30 a.m. overpass, and SW CRE value is from only 1:30 p.m. overpass. Thank you very

much for the reviewers' suggestions. Following the reviewers' comment and the method from L'Ecuyer et al. (2019), we recalculate the average shortwave fluxes as follows:

$$F_{daily} = F_{inst} \times \frac{Q_{daily}}{Q_{inst}} \qquad (1)$$

where $F_{daily}$ is the daily mean shortwave fluxes, $F_{inst}$ is the instantaneous fluxes from 2B-FLXHR-LIDAR, $Q_{daily}$ is the daily mean solar insolation, $Q_{inst}$ is the incoming solar of each profile from 2B-FLXHR-LIDAR. We recalculate all the results in Section 3.2. The new results are as follows:

| Region | TOA NCRE (W m⁻²) | | | BOA NCRE (W m⁻²) | | | ATM NCRE (W m⁻²) | | |
|---|---|---|---|---|---|---|---|---|---|
| | DCC | DCS | anvil | DCC | DCS | anvil | DCC | DCS | anvil |
| TP | -51.8 | -41.4 | -33.2 | -143.7 | -102.9 | -80.1 | 91.9 | 61.5 | 46.9 |
| TO | -80.8 | -13.6 | 4.7 | -226.7 | -115.0 | -83.1 | 145.9 | 101.3 | 87.8 |

**Table R1: The mean net cloud radiative effects (NCRE) of DCSs, DCCs and anvil in different regions. The estimated SW fluxes are normalized to the diurnally averaged insolation.**

| Region | TOA CRE (W m⁻²) | | | | | |
|---|---|---|---|---|---|---|
| | SWCRE | | | LWCRE | | |
| | DCC | DCS | anvil | DCC | DCS | anvil |
| TP | -205.3 | -153.5 | -124.1 | 152.3 | 110.7 | 89.4 |
| TO | --244.1 | -123.7 | -87.3 | 162.4 | 108.1 | 88.8 |
| | BOA CRE (W m⁻²) | | | | | |
| Region | SWCRE | | | LWCRE | | |
| | DCC | DCS | anvil | DCC | DCS | anvil |
| TP | -246.6 | -182.9 | -147.2 | 102.2 | 78.7 | 65.8 |
| TO | -262.1 | -133.1 | -94.4 | 35.5 | 18.7 | 13.1 |

**Table R2. The SW CRE (W m⁻²) and LW CRE (W m⁻²) at TOA and BOA of DCSs, DCCs and anvil in different regions. The estimated SW fluxes are normalized to the diurnally averaged insolation.**

[Figure]

**Figure R2: The proportion of each bin in the total sample of DCC, DCS and anvil in the TP (blue), TO (red) of the net CRE at the top of the atmosphere (TOA) (a, d, g), at the bottom of the atmosphere (BOA) (b, e, h) and in the atmosphere (ATM) (c, f, i). The dash lines show the peak of DCC. The estimated SW fluxes are normalized to the diurnally averaged insolation.**

[Figure]

**Figure R3: The same to Figure R2, but the results of the SW CRE.**

[Figure]

**Figure R4: The same as Figure R2, but the results of the LW CRE.**

[Figure]

**Figure R5: The mean vertical profiles of longwave (a, d), shortwave (b, e), and net (c, f) cloud radiative heating rates (CRH) (K d⁻¹) in the TP and TO. The red lines represent the results of DCCs and the blue lines represent the results of DCSs. The shadow range represents the standard deviation. The estimated SW fluxes are normalized to the diurnally averaged insolation.**

[Figure]

**Figure R6. The mean vertical profiles of cloudy (all-sky) heating rates (K d⁻¹) of DCCs over the TP, and TO. The shadow range represents the standard deviation. The estimated SW fluxes are normalized to the diurnally averaged insolation.**

[Figure]

**Figure R7. The same as Figure R6, but the results of clear-sky heating rates (K d⁻¹).**

This method roughly reduces the instantaneous observed SW flux to 40% of its original value. The relative magnitudes of the CRE of DCS, DCC, and anvils between the two regions remain unchanged. However, the absolute values of both the net CRE and the SW CRE have overall decreased. Due to the reduction in the absolute value of SW CRH after normalizing the SW fluxes, the net CRH no longer shows the vertical distribution similar to that of SW CRH but instead resembles that of LW CRH.

However, as mentioned in comment #12, twice-daily measurements cannot capture the full diurnal cycle of convective clouds. This method of normalizing the SW flux assumes that cloud cover remains constant throughout the day, as detected at 13:30 LT. In reality, the diurnal cycle of deep convection is significant. And individual DCSs have different life cycles. Therefore, we chose to present the instantaneous observation results of cloud radiative effects of DCSs in the main text. Additionally, we described the limitations of the twice-daily instantaneous

observations and the summary of the above results in the "Conclusions and discussion" section.

**See the lines:** 659-666.

16. *L315*: The sentence starting with "In particular" is the start of a new discussion on LW BOA CRE so I would suggest starting a new paragraph.

**Response:** Thank you very much for the reviewer's suggestion. It is revised according to the suggestion.

**See the line:** 364.

17. *L388*: What does "the high value of CRH, the difference between all-sky and clear-sky heating rates" mean?

**Response:** We mean that the CRH is calculated as the all-sky heating rates minus clear-sky heating rates. Therefore, we can separately analyze the heating rates under all-sky and clear-sky conditions to elucidate the reasons for the high values of CRH. The description in our manuscript may cause misunderstanding for readers. In the revised manuscript, we have changed the sentence to: "CRH is calculated as the all-sky heating rates minus clear-sky heating rates. Therefore, the reasons for the high value of CRH can be analysed through the results of heating rates under all-sky and clear-sky conditions."

**See the lines:** 442-444.

18. *L461*: What is the sample size of each bin for the different ACs in Figure 6?

**Response:** When analyzing the role of aerosols, we select the top 30% and bottom 30% of samples based on aerosol concentration. Each group is further divided into four bins. For each ACs group, the TP region has 17 samples per bin, while the TO region has 55 samples per bin. Limited sample sizes are unavoidable when studying cloud clusters rather than cloud profiles. Therefore, as noted in L461, the limited

sample size could potentially impact the results. Nevertheless, it is encouraging that despite the sample size of each bin is small, some results still demonstrate obvious correlations.

19. *L604-605*: Your reasoning would for the enhanced SW CRE of anvils over the Tibetan Plateau could be improved if you verify whether the cloud tops are more densely packed. Perhaps take a look at cloud optical depth, which can be found in the 2B-FLXHR-LIDAR data set.

**Response:** Thank you very much for the valuable suggestions from the reviewers. Cloud optical thickness indeed serves as a robust parameter for describing cloud density. Following the reviewer's suggestion, we have added the analysis of cloud optical thickness based on the parameters provided by 2B-FLXHR-LIDAR. The results are as follows:

[Figure]

**Figure R8 (same as the Fig. S4 in the revised manuscript). The histograms and mean values of cloud optical depth of anvils in different regions. The data is from 2B-FLXHR-LIDAR datasets.**

The results indicate that the cloud optical depth of anvils over the TP is larger, though the thickness is thinner. This suggests that anvils over the TP are more densely packed. Furthermore, this finding further supports our discovery of stronger shortwave radiative cooling of the anvils over the TP. These results have

been included in the supplement and corresponding descriptions have been added to the revised manuscript.

**See the lines:** 388-391.

**Technical Comments**

1. *L25-26*: "convective activity" instead of "convection activities"

   **Response:** It is corrected in the revised manuscript.

   **See the line:** 26-27.

2. *L32-32*: Modify this sentence to "Although deep convective cloud is less frequent compared to other cloud types, it has a more complicated vertical structure and larger extent, thus exerting a great influence on radiation and precipitation over the Tibetan Plateau region."

   **Response:** Thank you for the suggestion. It is corrected in the revised manuscript.

   **See the lines:** 33-35.

3. *L41*: Remove "Such as"

   **Response:** Revisions have been made according to the reviewer's suggestion.

   **See the line:** 42.

4. *L51-52*: "A complete deep convection system (DCS) should include both the deep convective core (DCC) and the anvils" is out of place and can be removed.

   **Response:** This sentence is removed.

5. *L110 & 151*: L'Ecuyer

**Response:** Sorry, this is a citation formatting error. It is corrected in the revised manuscript.

**See the lines:** 119 and 171.

6. *L233*: "Since the TO"

**Response:** It is revised according to the reviewer's suggestion.

**See the line:** 263.

7. *L239*: I think you mean DCC, not DCS

**Response:** Yes, the thickness of the DCC is used to characterize the thickness of the entire DCS in our study. To enhance readability, we revised the expression to "the DCCs over the TP are thinner".

**See the line:** 279.

8. *L250-251*: In the tropical region? Please rephrase.

**Response:** It is revised to "In the tropics".

**See the line:** 293.

9. *Fig S2-S3*: Are these mean values?

**Response:** Yes, the results in Fig. S2 and Fig. S3 are mean values. This explanation is added in the figure captions.

10. *L363*: By "larger", do you mean less negative?

**Response:** Yes, as the CRE is negative, "larger" means less negative. To improve readability, it is revised to "less negative".

**See the line:** 415.

11. *L366-367*: Are you comparing the anvil structure and radiative effects to the structure and radiative effects of the DCCs? Please clarify.

**Response:** Yes, we mean that compared with DCCs, the anvil clouds are thinner and higher. For this reason, the absolute value of SW CRE and LW CRE of DCSs, which is the average of the results of DCCs and anvils, is smaller than the results of DCCs. To enhance clarity, we have revised the sentence as follows: "Due to anvils being thinner and higher relative to DCCs, anvils reflect less solar radiation and emit less LW radiation. This results in the weaker CRE of DCSs compared to DCCs.".

**See the lines:** 418-419.

12. *L383*: Do you mean to say "below" instead of "within"?

**Response:** Yes, we mean the LW CRH is positive "below" 10 km. It is corrected in the revised manuscript.

**See the line:** 436.

13. *L385-386*: This is an incomplete sentence; please modify.

**Response:** It is revised to "This peak of LW CRH near the surface is more distinct in the TP compared with the TO, due to the colder boundary layer emitting less upwelling LW radiation in the TP (Haynes et al., 2013)."

**See the lines:** 438-440.

14. *L388*: Is "height bins with only several valid data are not shown" a typo? What is the value of "several valid data"?

**Response:** We only show the results of height bins with more than 50 samples. The results below 4.56 km are not shown. The value is added in the revised manuscript.

**See the line:** 441-442.

15. *L398*: Remove comma after LW CRH

   **Response:** It is corrected in the revised manuscript.

   **See the line:** 452.

16. *L463-465*: This sentence is incomplete.

   **Response:** Thank you for the reviewer's suggestion. The logical relationship between these two sentences is indeed somewhat confusing. The sentence "Although... reduced convection" is not closely related to the preceding or following text. To avoid confusing the readers, we have deleted and revised this part accordingly.

17. *L555*: Remove comma after While

   **Response:** It is corrected in the revised manuscript.

   **See the line:** 582.

**Reference**

Devasthale, A. and Fueglistaler, S.: A climatological perspective of deep convection penetrating the TTL during the Indian summer monsoon from the AVHRR and MODIS instruments, Atmos. Chem. Phys., 10, 4573-4582, https://doi.org/10.5194/acp-10-4573-2010, 2010.

Kottayil, A., Satheesan, K., John, V. O., and Antony, R.: Diurnal variation of deep convective clouds over Indian monsoon region and its association with rainfall, Atmospheric Research, 255, 105540, https://doi.org/10.1016/j.atmosres.2021.105540, 2021.

L'Ecuyer, T. S., Hang, Y., Matus, A. V., and Wang, Z.: Reassessing the Effect of Cloud Type on Earth's Energy Balance in the Age of Active Spaceborne Observations. Part I: Top of Atmosphere and Surface, J. Clim., 32, 6197-6217, https://doi.org/10.1175/JCLI-D-18-0753.1, 2019.

Randel, W. J., Park, M., Emmons, L., Kinnison, D., Bernath, P., Walker, K. A., Boone, C., and Pumphrey, H.: Asian Monsoon Transport of Pollution to the Stratosphere, Science, 328, 611-613, https://doi.org/10.1126/science.1182274, 2010.

Sherwood, S. C. and Wahrlich, R.: Observed Evolution of Tropical Deep Convective Events and Their Environment, Monthly Weather Review, 127, 1777-1795, https://doi.org/10.1175/1520-

0493(1999)127<1777:OEOTDC>2.0.CO;2, 1999.

Sun, Y., Wang, Y., Zhao, C., Zhou, Y., Yang, Y., Yang, X., Fan, H., Zhao, X., and Yang, J.: Vertical Dependency of Aerosol Impacts on Local Scale Convective Precipitation, J. Geophys. Res.-Atmos., 50, e2022GL102186, https://doi.org/10.1029/2022GL102186, 2023.

Xu, W. and Zipser, E. J.: Diurnal Variations of Precipitation, Deep Convection, and Lightning over and East of the Eastern Tibetan Plateau, Journal of Climate, 24, 448-465, https://doi.org/10.1175/2010JCLI3719.1, 2011.